

# Spatio-temporal patterns and trends of streamflow in water-scarce Mediterranean basins

Laia Estrada[1,2], Xavier Garcia[1,2], Joan Saló[1,2], Rafael Marcé[1,2,3], Antoni Munné[4], Vicenç Acuña[1,2]

[1]Catalan Institute for Water Research (ICRA - CERCA). Carrer Emili Grahit 101, 17003 Girona (Spain)
[2]University of Girona. Plaça de Sant Domènec 3, 17004 Girona (Spain)
[3]Current address: Integrative Freshwater Ecology, Centre for Advanced Studies of Blanes (CEAB-CSIC), Blanes (Spain)
[4]Catalan Water Agency (ACA). Carrer Provença 260, 08008 Barcelona (Spain)

*Correspondence to*: Laia Estrada (lestrada@icra.cat)

**Abstract.** The issue of water scarcity, exacerbated by climate change and demographic increase, has become a growing concern in many regions throughout the world. Understanding hydrological behaviour to promote resilient and sustainable water management is paramount. Hydrological models that integrate natural processes and anthropogenic alterations of the basin's hydrology are a powerful tool to support decision-making. We developed a SWAT+ hydrological model including stakeholder expert knowledge on water management and introducing a novel calibration and validation approach suitable for

heterogeneous basins in space and / or time. We also assessed spatio-temporal patterns and trends of streamflow during the first two decades of the 21[st] century in the Catalan River Basin District, in the western Mediterranean, using a wide variety of indicators to fully characterize the hydrological regime. We calibrated and validated the model using data from 50 gauging stations, verifying the usefulness of the new calibration and validation strategy. Co-development with stakeholders and the integration of expert knowledge, most notably on reservoir operations, helped improve model performance. Results revealed

a generalized streamflow reduction, as well as increased dominance of streamflow flashiness and zero-flows recurrence. We also observed differences in seasonal trends, with autumn being the most affected season. These results provide insights into how climate change and anthropogenic pressures are going to keep affecting water resources availability in the future, thus raising the need for sustainable management practices in the Catalan River Basin District, as well as other regions vulnerable to water scarcity.

**1 Introduction**

Water scarcity, a situation in which freshwater demands exceed its availability due to climatic or non-climatic factors, is a growing concern in many regions, with half the world's population being subject to severe water scarcity (Caretta et al., 2022). The Mediterranean region is already prone to water scarcity due to its intrinsic high inter-annual rainfall variability, but rising demands and climate change, which includes warming at faster rates than the global mean as well as reduced

rainfall, will exacerbate this issue (Cramer et al., 2018).





Promoting sustainable water resources management is thus of great importance to address the challenges brought on by the combined effects of climate change, demographic growth, and overexploitation of resources (Zribi et al., 2020). Using hydrological models to predict water resources availability under different climate and/or management scenarios is thus necessary to identify efficient adaptation measures and support decision-making (Loucks and van Beek, 2017).

Despite the potential contribution of hydrological models to the water management sector, their practical implementation remains a challenge (Pezij et al., 2019). In some cases, the main issue is the lack of active involvement from end-users (i.e., stakeholders and water managers) during the model's implementation process, which can lead to the actual management needs not being addressed by the model, or the model's capabilities not being effectively conveyed (Loucks and van Beek, 2017). Co-development with end-users can significantly improve the model through the inclusion of first-hand expert

knowledge (e.g., addressing specific needs or simulating more accurate management operations). Additionally, this collaborative approach increases the likelihood of the model being used in the planning and management processes, as end-users become aware of its capabilities and limitations (Højberg et al., 2013; Bots et al., 2011).

Many studies have successfully applied hydrological modelling in watersheds susceptible to water scarcity to characterize streamflow dynamics and evolution (e.g., Swain et al., 2020; Tanner et al., 2022), including in the Mediterranean region

(e.g., Brouziyne et al., 2021; De Girolamo et al., 2022). While many studies focus on basins with natural or near-natural flow regimes, it is important to note that larger regulated basins, typically more relevant to water management, are often excluded. Some studies include regulated basins with coupled hydrological-water allocation models (Haro-Monteagudo et al., 2020) or a specific reservoir module (Eekhout et al., 2020). However, these studies perform separate calibrations for the hydrological and the water allocation/reservoir models. Moreover, hydrological calibration is either performed upstream of the reservoirs

or using naturalized streamflow data. To our knowledge, there are no studies yet involving an integrated model (i.e., a single model including both hydrological and water allocation processes) calibrated in a single process using both natural and altered flow regimes.

In relation to the calibration process, most model applications rely on streamflow data from one or few gauging stations within the basin. This practice, however, constrains the accuracy and practicality of the calibrated model, especially in larger

case studies with heterogeneous rainfall regimes, varying elevation, and diverse land uses (i.e., where spatial and often temporal variability are pronounced). Despite the importance of multi-site calibration for large and/or highly heterogeneous basins being noted, most studies use a limited number of gauging stations, most likely due to data availability (e.g., Pandey et al., 2020; Wang et al., 2012; Wi et al., 2015).

Moreover, the division of data into calibration and validation periods can significantly affect model results. This highlights

the importance of selectively choosing those periods, despite the fact that in most studies it is done arbitrarily (Myers et al., 2021), allocating either the first or last recorded years for calibration, and the rest for validation. Although generally it is recommended that both periods are statistically similar (Abbaspour et al., 2018; Dakhlaoui et al., 2019), using distinctly different periods can also be interesting depending on the intended use of the calibrated model. Specifically, when the model is to be applied to future climate and/or land use change scenarios (Daggupati et al., 2015). An alternate approach is to



randomize the data splitting to obtain a calibrated model that can be accurately used under conditions that are both similar and different from those of the calibration period, as well as to ensure that the model performance is not biased by the allocation of data (Daggupati et al., 2015; Bennett et al., 2013). This is of particular interest for watersheds with spatio-temporally heterogeneous gauging data, as each individual record presents different lengths and hydrological characteristics. Selectively choosing the calibration and validation period for each gauging station may be challenging, but the random

approach effectively ensures that all spatio-temporal variability is captured across all stations provided that enough data is available. However, this has not been explored before in large-scale applications with many gauging stations, as generally only calibration is performed (Chawanda et al., 2020), or the last years of the record are arbitrarily chosen for validation without considering the statistical similarity (or dissimilarity) between the two periods (Abbaspour et al., 2015; Piniewski and Okruszko, 2011).

As previously stressed, hydrological models can be used to support water managers, but also to identify and analyse streamflow patterns and trends in ungauged basins, which is of great interest to understand the evolution of hydrologic dynamics under both natural and anthropogenic pressures. However, many studies addressing this topic only use observed flows, which can constrain the spatial interpretation of trends. Hydrological models, on the other hand, can bridge this gap by providing spatially coherent patterns and trends of streamflow (Stahl et al., 2012). At the global level, Gudmundsson et al.

(2019) used streamflow observations to identify trends in low, mean and high flows, and Gudmundsson et al. (2021) verified that observed trends were only reproducible with models if anthropogenic climate change was considered. Both observations and models show a strong and significant decreasing trend of streamflow in the Mediterranean region, while the opposite is observed in northern Europe. This fact is also confirmed in studies at the European level using observations (Blöschl et al., 2019; Masseroni et al., 2021; Stahl et al., 2010) and models (Gudmundsson et al., 2017; Stahl et al., 2012). Local studies

have also identified this decreasing trend in the Mediterranean using observations (Folton et al., 2019; Lutz et al., 2016) and models (Llanos-Paez et al., 2023). Many of these studies only focus on trends in annual streamflow, with some also evaluating trends on low, high, and monthly flows. However, none delve into trends informing of other key hydrological regime characteristics than magnitude (i.e., timing, duration, frequency, and rate of change), excluding Folton et al. (2019) and Llanos-Paez et al. (2023), where trends on the duration and timing of low flows and on the duration, timing and

frequency of zero-flows are evaluated, respectively.

The objective of this research is to develop a hydrological model that can serve as a valuable tool in the water management sector by integrating both physical and anthropogenic processes, including expert knowledge of management operations. We also introduce a novel calibration and validation strategy, which overcomes the limitations of conventional approaches, particularly when dealing with relatively large modelling studies involving numerous and spatio-temporal heterogenous

gauging stations. Moreover, this new approach can deal with records differing in length and/or with significant gaps, as it determines calibration and validation periods for each gauging station individually. The widely employed hydrological modelling Soil Water and Assessment Tool (SWAT) is used, more particularly its revised version SWAT+ (Bieger et al., 2017). Another objective is to identify and characterize spatio-temporal patterns and trends of streamflow in Mediterranean



basins during the first two decades of the 21st century. This research provides valuable insights into modelling improvements

achieved by promoting co-development with end-users and introducing a novel calibration and validation approach. It also

contributes to deepen our understanding of how water-scarce Mediterranean basins behave under changing climatic

conditions and anthropogenic pressures.

## 2 Material and methods

### 2.1 Study site

The study was performed in the Catalan River Basin District (CRBD), located in the northeastern part of the Iberian

Peninsula (western Mediterranean). It covers an area of over 16,000 km$^2$, comprising of several small/medium-sized river

basins, and it is managed as a single River Basin District by the Catalan Water Agency. However, for this study only part of

the CRBD is considered, including the Llobregat, Ter, Fluvià, Besòs, Tordera and Foix basins (Figure 1). This selection was

based on data availability and the overall significance of these basins, as they host most of the water resources and demand

in the region.

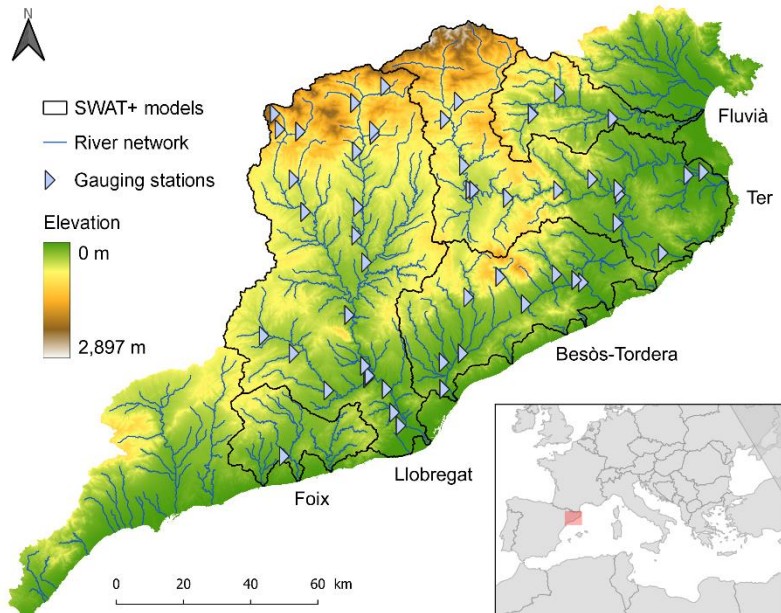

**Figure 1: Location of the CRBD in the Western Mediterranean. The study area has been divided into five SWAT+ models, named after its main river basin.**

The elevation within the CRBD ranges from 0 to 2910 m a.s.l. Almost 60% of the area is covered by forests, mainly

coniferous, although in the eastern basins broad-leaved forests are dominant, especially sclerophyll forests. Crops are mainly

found in the medium and lower parts of the basins, the most common being barley, wheat, and grape vines. The majority is

rain-fed, with only 3.2% irrigated.





For the period 2001-2022, the CRBD received on average 660 mm of annual precipitation. However, as is characteristic of the Mediterranean climate, there is a high interannual variability, ranging between 480 and 990 mm. There is also a high

spatial variability within the CRBD, with annual averages ranging from 900 mm in the Pyrenees to 450 mm in coastal regions.

Water demand within the CRBD exceeds 1000 hm$^3$/year, of which 42 % is for agricultural use, 32 % for domestic use, 10 % for industrial use, and the remaining 15 % for other urban uses (ACA, 2021). Over 7.1 million people live in the CRBD, of which 91 % live within the basins chosen for this study.

## 2.2 Hydrological modelling with SWAT+

SWAT is a semi-distributed and physically-based ecohydrological model widely used worldwide (Abbaspour et al., 2015; Gassman et al., 2014; Samimi et al., 2020), including many applications in Mediterranean basins (Boithias et al., 2017; Brouziyne et al., 2021; De Girolamo et al., 2022). The upgraded SWAT+ offers an improved spatial representation of hydrological processes and interactions within a watershed (Bieger et al., 2017, 2019), as well as more flexibility in defining

management schedules and operations using decision tables (Arnold et al., 2018). SWAT+ has also been successfully used in Mediterranean basins (Pulighe et al., 2021; Castellanos-Osorio et al., 2023; Llanos-Paez et al., 2023).

SWAT+ divides the basin into several sub-basins, which can be further divided into landscape units (LSUs: floodplain and upland), and those into hydrological response units (HRUs), the smallest unit at which hydrological processes are simulated. Each HRU consists of the area within a LSU with a unique combination of land use, soil type and topographic slope (Bieger

et al., 2017). Connectivity between spatial objects, including HRUs and LSUs as well as channels, aquifers and reservoirs, can be adjusted by the user to realistically represent the basin's physical characteristics (Bieger et al., 2019).

### 2.3 Co-development with end-users

We involved end-users (i.e., water managers from the Catalan Water Agency) in the development and implementation of the SWAT+ model for the CRBD. This co-development led to the inclusion of valuable expert knowledge on actual

management practices in the CRBD, resulting in a more accurate hydrological model. Notably, we incorporated expert knowledge on actual reservoir release operations into custom-built decision tables. We also included expert knowledge on irrigation, inter-basin water transfers, urban abstractions and point source discharges.

We conducted several meetings to discuss progress on model development, as well as SWAT+ training sessions, in order to help end-users familiarize themselves with the model and its capabilities as a tool to support decision making.

### 2.4 SWAT+CRBD inputs and configuration

The basic inputs needed to set up a SWAT+ model are a digital elevation model (DEM), land use map, soil map, and weather data. The weather data consists of daily measurements of precipitation, maximum and minimum temperatures, relative humidity, wind speed and solar radiation, as well as a database of monthly statistics to fill in null values. A



floodplain map is also required if the LSUs floodplain and upland within a sub-basin are to be separated. The basic inputs
used for the SWAT+CRBD model are listed in Table 1. Additional input data, including reservoir operations, cropland
management, and point source discharges and abstractions, were also used to enhance the SWAT+CRBD model.

**Table 1. SWAT+ input data, resolution, and sources. Weather data include precipitation, maximum and minimum temperature, relative humidity, wind speed, and solar radiation.**

| SWAT+ inputs | Resolution | Source |
|---|---|---|
| Digital Elevation Model | 70 m | Cartographic and Geological Institute of Catalonia. Modified to improve stream delineation and adjust resolution. https://www.icgc.cat/en/Downloads/Elevations https://www.icgc.cat/en/Downloads/Elevations |
| Land use map | 100 m | CORINE Land Cover 2018. https://land.copernicus.eu/pan-european/corine-land-cover/clc2018 |
|  | 1 m | Land use map of Catalonia 2018, Cartographic and Geological Institute of Catalonia. https://www.icgc.cat/Descarregues/Mapes-en-format-d-imatge/Cobertes-del-sol |
| Soil map | 250 m | Digital SoiL OpenLand Map, WaterITech. https://www.wateritech.com/data |
| Floodplain map | 20 m | Floodable area for T=10y, Catalan Water Agency. https://sig.gencat.cat/visors/VISOR_ACA.html |
| Weather data | 1 day | Meteorological Service of Catalonia (Meteocat) https://www.meteo.cat |
|  | Monthly statistics | Spain weather generator database for SWAT+. https://swat.tamu.edu/data/spain/ |

### 2.4.1 Digital elevation model

The original DEM had a resolution of 15 m but was modified to improve the delineation of the river network through plain
terrain, as well as resampled to a 70 m resolution. The lower resolution is more reasonable to the extent of the CRBD. A
threshold of 5 km$^2$ of drainage area was used for the channel delineation, however the river network was then manually
modified by merging or splitting default river reaches generated by the automatic delineation. The criteria used were the
official body masses defined by the Catalan Water Agency, as well as other constraints (i.e., gauging stations, WWTP and
industrial point source discharges).

### 2.4.2 Land use data

The land use map was based on the CORINE Land Cover (CLC) 2018 (EEA, 2018) and adapted to SWAT+ nomenclature.
We also used the land cover map of Catalonia (ICGC, 2018) to differentiate between deciduous and sclerophyll broad-leaved
forests, as well as more specific crop land uses.

### 2.4.3 Soil data

We used the Digital SoiL OpenLand Map (DSOLMap), a newly developed soil map at 250 m resolution, with a detailed six-
horizons soil profile, specifically tailored to use with the SWAT+ model (López-Ballesteros et al., 2023).



### 2.4.4 Floodplain map

We used a floodplain map to differentiate between the LSUs floodplain and upland within a subbasin. We used the floodable area for a return period of 10 years, and we complemented it by using geomorphological criteria in the sections of the river network where this area was not defined.

### 2.4.5 Weather data

We have used daily data from 141 meteorological stations spread out across the CRBD for the simulation period 2000-2022. We used a one-year warmup, as it is the SWAT+ default. Most of the data was provided by the Catalan Water Agency, who previously gathered data from different sources, including the Meteorological Service of Catalonia (SMC), the Spanish Meteorological Agency (AEMET), and the Ebro Basin Water Authority (CHE). Data for the years 2021-2022 is exclusively from the SMC (Table 1).

Not all stations have complete records for the period 2000-2022, therefore, the Spain weather generator dataset for SWAT+ was used to fill daily missing values with monthly statistics (Senent-Aparicio et al., 2021).

### 2.4.6 Additional data

We included five reservoirs in the SWAT+CRBD model. Each reservoir has its own decision table where the release operations are defined according to actual release rules provided by the Catalan Water Agency. There are five possible release scenarios depending on the reservoir level: water excess, normality, drought alert, exceptional drought, and emergency. In the first case, the reservoir is full and all daily inputs through the river network are then released. During normality, a fixed daily release rate for every month is defined according to the mean releases from the last five years. For the rest of the scenarios, the fixed release rate is decreased according to the drought restrictions defined by the Catalan Water Agency. When possible, the performance of the decision tables was tested in R using real daily inputs and water level, and the estimated fixed release rates for every scenario/month were calibrated to the median value instead.

WWTP and industrial discharges were included in the SWAT+CRBD model by means of point source objects to account for the added volumes to the river network, as in some smaller rivers their influence is considerable. Abstractions for urban use were also accounted for with negative point sources.

We have implemented decision tables to manage crop schedules and operations based on plant growth and soil water content for most of the crops, except for orchards, grape vines, and olive trees, which have a manual harvest operation. Irrigation is also managed by decision tables as well as the water allocation routines native to SWAT+. Irrigated HRUs are grouped into irrigation districts with its own irrigation source (either a channel, a reservoir, or an aquifer) and a threshold to only allow water abstraction if this is exceeded (respectively, environmental flow, minimum volume, and minimum water table). On each simulation day, if water stress in a HRU exceeds a threshold, irrigation demand is defined. We modified the SWAT+ source code so that the irrigation demand is the difference between the potential evapotranspiration and the real





evapotranspiration instead of a fixed volume. If an HRU has an irrigation demand and there is enough water in the irrigation

200    district's source, then this HRU is irrigated.

## 2.5 Sensitivity analysis

We performed a global sensitivity analysis using the variance-based Fourier amplitude sensitivity test (FAST) method (Cukier et al., 1973) using the R packages Fast (Fast: Implementation of the Fourier Amplitude Sensitivity Test (FAST)) and SWATplusR (Schürz, 2022). Variance-based methods are widely used for parameter sensitivity analysis in hydrological

205    models (Song et al., 2015), including SWAT (Guse et al., 2014) and SWAT+ (Llanos-Paez et al., 2023) applications.

For the Llobregat and Ter models, we performed the sensitivity analysis with 30 parameters (Table 2), while for the rest of the SWAT+ models we used 28 as the last two parameters are related to reservoirs, and these models either do not include reservoirs (Besòs-Todera, Fluvià) or gauging stations are located upstream (Foix). We ran 5763 simulations for the Llobregat and Ter models and 4795 for the rest, as the number of iterations required to determine the sensitivity indices is related to the

210    number of parameters. We used the objective function Kling-Gupta Efficiency (KGE) (Gupta et al., 2009) and considered a parameter sensitive if their partial variance accounted for more than 1 ‰ of the total (Table 2).

**Table 2. Parameters used in the sensitivity analysis. For 'pctchg' the parameter is altered by a percentage of the default value, while for 'absval' the absolute value is replaced. For each sub-model, bolded variances indicate that a parameter is sensitive. Llob.: Llobregat; B-T: Besòs-Tordera.**

| Parameter | Description | Change | Range | | FAST variance (‰) | | | | |
|---|---|---|---|---|---|---|---|---|---|
| | | | Min | Max | Llob. | Ter | B-T | Fluvià | Foix |
| cn2 | Condition II curve number | pctchg | -50 | 50 | **786.2** | **815.8** | **899.1** | **755.1** | **950.9** |
| cn3_swf | Soil water adjustment factor for CN3 | absval | 0 | 1 | **3.0** | **59.9** | **30.5** | **34.9** | **14.3** |
| ovn | Overland flow Manning's n | pctchg | -50 | 50 | <1 | <1 | <1 | <1 | <1 |
| lat_ttime | Lateral flow travel time (days) | absval | 0.5 | 180 | **2.4** | <1 | **1.9** | **3.8** | <1 |
| latq_co | Lateral flow coefficient | absval | 0 | 1 | **5.0** | **20.8** | **3.0*** | **25.5** | <1 |
| canmx | Maximum canopy storage (mm) | absval | 0 | 100 | **116.3** | **2.3** | **18.6** | **5.1** | **24.5** |
| esco | Soil evaporation compensation factor | absval | 0 | 1 | <1 | <1 | <1 | **1.0** | <1 |
| epco | Plant uptake compensation factor | absval | 0 | 1 | <1 | **1.4*** | **1.3*** | **1.3*** | <1 |
| perco | Percolation coefficient | absval | 0 | 1 | **1.1** | **2.1** | <1 | **5.8** | <1 |
| z | Depth from soil surface to bottom of layer (mm) | pctchg | -50 | 50 | **7.0** | **19.4** | **1.5** | **30.3** | <1 |
| bd | Moist bulk density of soil layer (g/cm3) | pctchg | -50 | 50 | **29.9** | **38.9** | **26.4** | **84.4** | **1.3** |
| awc | Available water capacity of soil layer (mm H2O/mm soil) | pctchg | -50 | 50 | **2.1** | **2.7*** | **4.0** | <1 | **2.7** |
| k | Saturated hydraulic conductivity of soil layer (mm/hr) | pctchg | -50 | 50 | <1 | **1.1** | <1 | **3.3** | <1 |
| plaps | Precipitation lapse rate (mm/km) | absval | 0 | 200 | <1 | <1 | <1 | <1 | <1 |
| tlaps | Temperature lapse rate (deg C/km) | absval | -10 | 10 | <1 | <1 | <1 | <1 | <1 |
| surlag | Surface runoff lag coefficient | absval | 0.05 | 24 | <1 | <1 | <1 | <1 | <1 |
| evrch | Reach evaporation adjustment factor | absval | 0.5 | 1 | **1.2** | <1 | <1 | <1 | <1 |





| evlai | Leaf area index at which no evaporation occurs from water surface | absval | 0 | 10 | <1 | <1 | <1 | <1 | <1 |
|---|---|---|---|---|---|---|---|---|---|
| ffcb | Initial soil water storage (fraction) | absval | 0 | 1 | **1.4** | <1 | <1 | <1 | <1 |
| chn | Channel Manning's n | absval | 0 | 0.3 | **5.9*** | **1.5** | **7.8*** | **2.0*** | <1 |
| chk | Channel bottom conductivity (mm/day) | absval | 0 | 1 | **41.9** | **34.7** | **15.9** | **46.6** | **4.7** |
| k_res | Hydraulic conductivity of the reservoir bottom (mm/hr) | absval | 0 | 1 | <1 | <1 | - | - | - |
| evrsv | Lake evaporation coefficient | absval | 0 | 500 | <1 | <1 | - | - | - |
| alpha | Alpha factor for groundwater recession curve (1/days) | absval | 0 | 1 | <1 | <1 | <1 | <1 | <1 |
| bf_max | Baseflow rate when the entire area is contributing to baseflow (mm) | absval | 0.1 | 2 | <1 | <1 | <1 | <1 | <1 |
| deep_seep | Recharge to deep aquifer (fraction) | absval | 0.001 | 0.4 | <1 | <1 | <1 | <1 | <1 |
| sp_yld | Specific yield for shallow aquifer (m3/m3) | absval | 0 | 0.5 | <1 | <1 | <1 | <1 | <1 |
| flo_min | Water table depth for return flow to occur (m) | absval | 0 | 50 | <1 | <1 | <1 | <1 | <1 |
| revap_co | Groundwater revap coefficient (fraction) | absval | 0.02 | 0.2 | <1 | <1 | <1 | <1 | <1 |
| revap_min | Water table depth for revap to occur (m) | absval | 0 | 50 | <1 | <1 | <1 | <1 | <1 |

* Variance assessed for monthly streamflow. These parameters are considered sensitive even though the daily variance is less than 1‰

## 2.6 Calibration and validation

We conducted the model calibration with daily streamflow data obtained from 50 gauging stations distributed across the study area. We calibrated each of the five SWAT+ models independently, respectively using 23, 14, 9, 3 and 1 gauging station for the Llobregat, Ter, Besòs-Tordera, Fluvià, and Foix models (Figure 1). The simulation period is from 2001 to 2022, however, many gauging stations present significant gaps in their records. These gaps in data are characterized by variability in terms of record length and temporal distribution throughout the simulation period. Traditional methods to determine calibration and validation periods involve either discretionarily using a period at the beginning or end for validation and the rest for calibration or selecting statistically similar periods. However, due to the spatial and temporal heterogeneity in our gauged data, these methods are not suitable for our study. To address this, we randomly selected the calibration and validation periods for each station independently. Specifically, we allocated 70% of the records for calibration and reserved 30% for validation. To prevent numerous alternating periods, we introduced the constraint of a single validation window, which can randomly occur at any point within the station's record. Consequently, this approach resulted in one or two calibration periods and one validation period. We assume that with the random approach all spatiotemporal variability is captured. Besides, all bias that might arise during the selection is removed.

For each SWAT+ model and its corresponding sensitive parameters (Table 2), we used Latin hypercube sampling to generate 2000 parameters combinations. Only one calibration iteration was performed, as during the sensitivity analysis we already ran many simulations and further iterations were not expected to yield better results. The model performance was assessed by the objective functions KGE and percent bias (PBIAS). We considered that results were "satisfactory" if KGE > 0.5 and -





25% < PBIAS < 25%, "good" if KGE > 0.65 and -15% < PBIAS < 15%, and "very good" if KGE > 0.75 and -10% < PBIAS < 10% (Moriasi et al., 2007; Odusanya et al., 2021).

**2.7 Data analysis**

To identify and characterize spatio-temporal patterns and trends of streamflow in the CRBD, we calculated 40 annual hydrological indicators for each of the 999 SWAT+CRBD river reaches, as well as the annual percentage of reaches that dry at least once for the whole river network (Table 3). We used indicators characterizing low, medium and high flows in relation of their magnitude, frequency, duration, timing and rate of change (Richter et al., 1996; Alcaraz-Hernández et al., 2023), as well as several zero-flow indicators to characterize flow intermittency (Llanos-Paez et al., 2023).

**Table 3: Hydrological indicators used for the analysis of flow patterns and trends.**

| Hydrological indicator | Units | Regime characteristics |
|---|---|---|
| Medium flows | | |
| Median of daily flow for each month | m3/s | Magnitude, timing |
| Median of annual daily flow | m3/s | Magnitude |
| High flows | | |
| 90th percentile of annual daily flow | m3/s | Magnitude |
| Annual maximum daily flow | m3/s | Magnitude, duration |
| 3, 7, 30 and 90-day means of maximum daily flow | m3/s | Magnitude, duration |
| Julian date of maximum daily flow | Julian day | Timing |
| Number of high pulses | Events/year | Frequency |
| Mean duration of high pulses | Days | Duration |
| Low flows | | |
| 10th percentile of annual daily flow | m3/s | Magnitude |
| Annual minimum daily flow | m3/s | Magnitude, duration |
| 3, 7, 30 and 90-day means of minimum daily flow | m3/s | Magnitude, duration |
| Julian date of minimum daily flow | Julian day | Timing |
| Number of low pulses | Events/year | Frequency |
| Mean duration of low pulses | Days | Duration |
| Zero-flows | | |
| Total number of days with zero-flow | Days/year | Duration |
| Number of periods of consecutive zero-flow days | Events/year | Frequency |
| Mean duration of zero-flow events | Days | Duration |
| Julian date of first zero-flow event | Julian day | Timing |
| Median julian date of zero-flow events | Julian day | Timing |
| Percentage of river network that dries at least once per year | % | Frequency |
| General | | |
| Rise rate: means of all positive differences between consecutive days | m3/s | Magnitude, rate of change |
| Fall rate: mean of all negative differences between consecutive days | m3/s | Magnitude, rate of change |





| Number of flow reversals | Reversals/year | Frequency, rate of change |
|---|---|---|
| Sum of all annual flow | hm3 | Magnitude |

We applied a modified Mann-Kendall test (Yue and Wang, 2004) to identify statistically significant temporal trends with a *p-value* lower than 0.05 while accounting for serial autocorrelation. The Theil-Sen estimator (Theil, 1950; Sen, 1968),
hereby referred to as only Sen's slope, was employed to assess the magnitude of trends. Sen's slopes for indicators with units of flow ($m^3/s$) were standardized to remove the stronger influence of river segments with higher streamflow, and thus avoid a skewed interpretation of spatial patterns. Both methods have been widely used in hydrology to identify and quantify trends, as demonstrated in prior studies (e.g., Clavera-Gispert et al., 2023; Gudmundsson et al., 2017; Khorchani et al., 2021).

## 3 Results

### 3.1 Sensitivity analysis

The FAST sensitivity analysis identified in each SWAT+ model between 6 and 13 sensitive parameters (Table 2). Most of the sensitive parameters are identified in multiple models, and they are mostly related to surface and plant processes (cn2, cn3_swf, canmax, epco), soil processes (latt_ttime, lat_co, esco, perco, z, bd, awc, k) and channel processes (chn, chk).

### 3.2 Calibration and validation

The comparisons between observed and simulated daily streamflow for representative gauging stations of the main rivers of the CRBD demonstrate good model performance (Figure 2), and the values of the objective functions overall indicate satisfactory results for both the calibration and validation periods (Table 4). For the validation period, KGE values are satisfactory or above both for the daily and monthly timestep, and PBIAS values are good or very good.

**Table 4. Kling-Gupta Efficiency (KGE) and percent bias (PBIAS) results for each of the five SWAT+ models that configure**
**SWAT+CRBD, assessed at a daily and monthly timestep for the calibration and validation period.**

| SWAT+ model | Daily timestep | | | | Monthly timestep | | | |
|---|---|---|---|---|---|---|---|---|
| | Calibration | | Validation | | Calibration | | Validation | |
| | KGE | PBIAS | KGE | PBIAS | KGE | PBIAS | KGE | PBIAS |
| Llobregat | 0.46 | -19.1 | 0.53 | -14.5 | 0.63 | -19.3 | 0.68 | -14.6 |
| Ter | 0.48 | -8.7 | 0.51 | -10.7 | 0.77 | -9.2 | 0.82 | -10.6 |
| Besòs-Tordera | 0.66 | 0.5 | 0.70 | -5.1 | 0.87 | 0.3 | 0.85 | -4.9 |
| Fluvià | 0.53 | -12.6 | 0.59 | -14.9 | 0.59 | -12.6 | 0.58 | -14.6 |
| Foix | 0.55 | -5.1 | 0.74 | -13.2 | 0.76 | -3.9 | 0.71 | -13 |







**Figure 2: Observed and simulated daily streamflow for the period 2001-2022 in the six main rivers of SWAT+CRBD. Individual KGE and PBIAS values for both the calibration and validation periods are also shown.**



### 3.3 Analysis of spatial and temporal patterns

The analysis of the simulated streamflow for the period 2001-2022 in the CRBD revealed several spatio-temporal patterns and trends (Figures 3-7). The advantage of using simulated streamflow instead of observed is working with 999 values for each indicator instead of only 50 (gauging stations), which allows us to better observe spatial patterns. Moreover, some gauging stations present gaps for the period 2001-2022, so simulated streamflow also provides a complete temporal series.

The evolution of the percent of river segments that dry at least once a year for the period 2001-2022 indicates a drying

tendency in the CRBD (Figure 3a), which can be positively correlated to an increase in mean annual temperature. While individual annual percentages correlate directly with mean annual rainfall, there is no apparent decreasing trend in the latter. We observe a general decrease in total annual flow, except for the headwaters of the Llobregat basin (Figure 3b). However, this region shows a poorer model adjustment (see discussion on section 4.1), which compromises the reliability of this result.

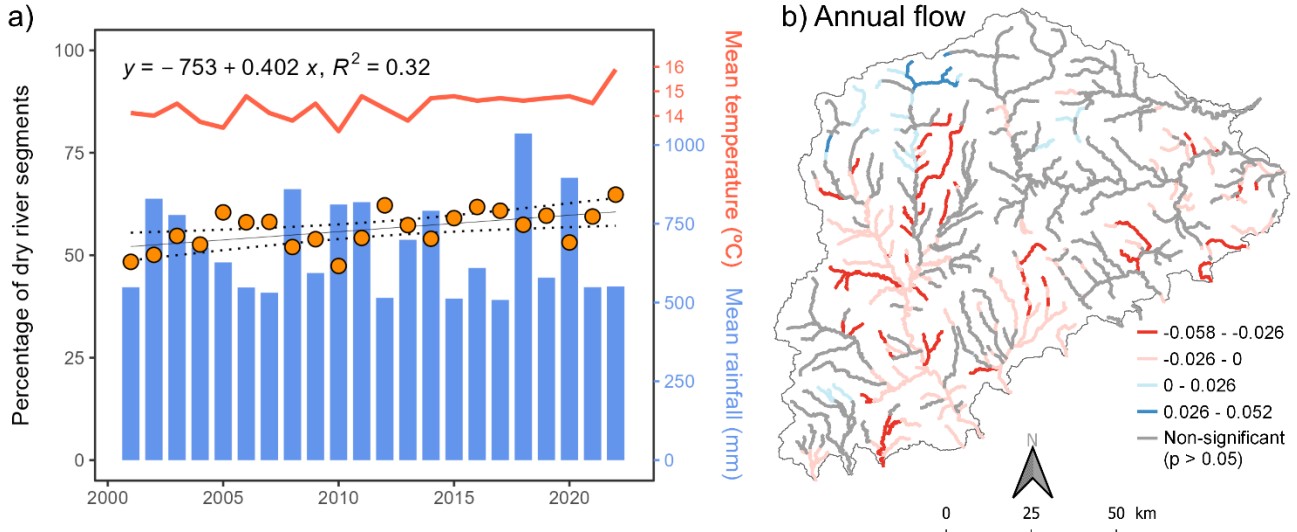

**Figure 3: a) Evolution of annual percentage of river segments that dry at least once a year and mean annual rainfall and temperature. b) Spatial distribution of standardized Sen's slope for the hydrological indicator total annual flow.**

Significant trends in median flow are most generally negative, both in annual (Figure 4a) and seasonal (Figure 4b-e) median flows, being the Llobregat basin the most clearly affected. Autumn is the season with the most widespread negative trends (4e). Positive trends are not as dominant and are generally located in the headwaters, while lower regions of the Ter basin

also show positive trends during winter (Figure 4b).



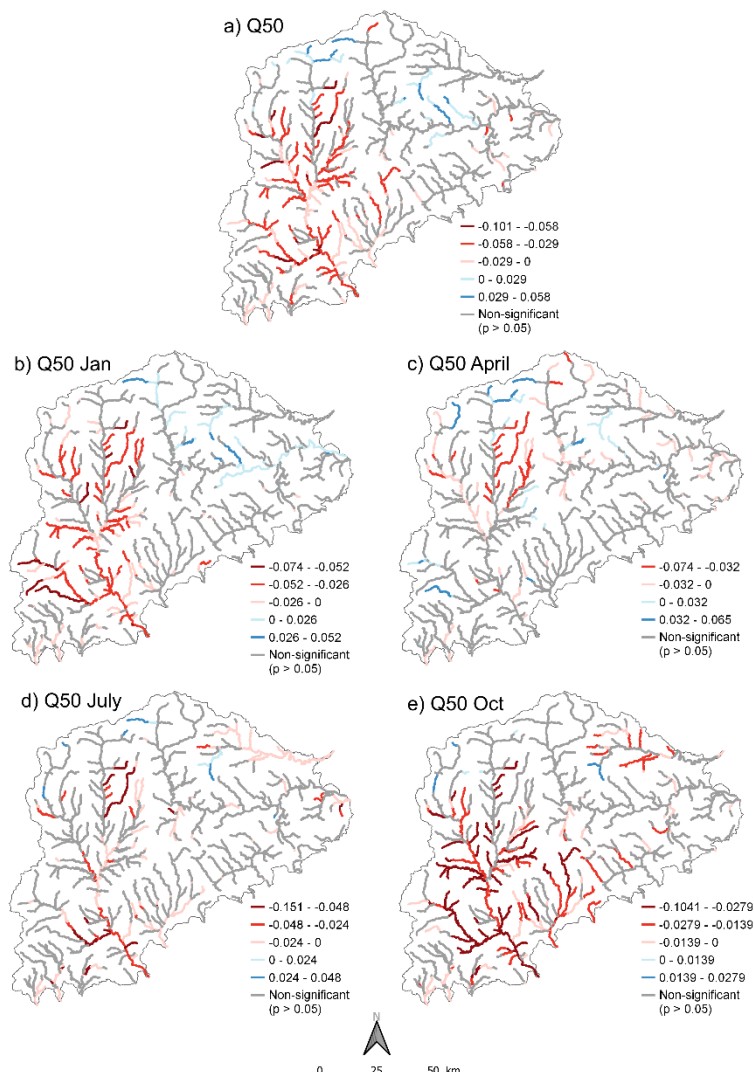

**Figure 4: Spatial distribution of standardized Sen's slope for the hydrological indicators annual Q50 (a) and Q50 in January (b), in April (c), in July (d), and in October (e), representative of the different seasonal flow patterns.**

Negative trends in the magnitude of high flows (90th percentile) are also consistently dominant, although some significant
positive trends can also be found mainly in the headwaters of the Llobregat basin (see discussion on section 4.1), as well as some other river segments in the Ter and Fluvià basins (Figure 5a). However, for low flows (10th percentile) most river segments do not present significant trends (Figure 5b), primarily because the 10th percentile threshold is zero in many of these segments. The few significant trends are mostly negative and located in the lower sections of the Llobregat basin and the Besòs basin. Conversely, positive trends can be observed in the headwaters of the Fluvià basin and downstream of the
reservoirs in the Ter basin.




Concerning the frequency of high and low-flow events, we observe a distinct spatial pattern for the number of high-flow events per year, with positive trends in the north (Pyrenees headwaters) and negative trends in the southern/coastal regions (Figure 5c). Trends for the number of low-flow events are generally non-significant, but segments with significant trends show an increase of annual episodes (Figure 5d). Mean duration of high-flow events generally decreases across the entire

study area (Figure 5e), while significant trends for mean duration of low-flow events show a decrease in the Pyrenees/northern regions and an increase in the Besòs, Foix and lower Llobregat basins (Figure 5f).

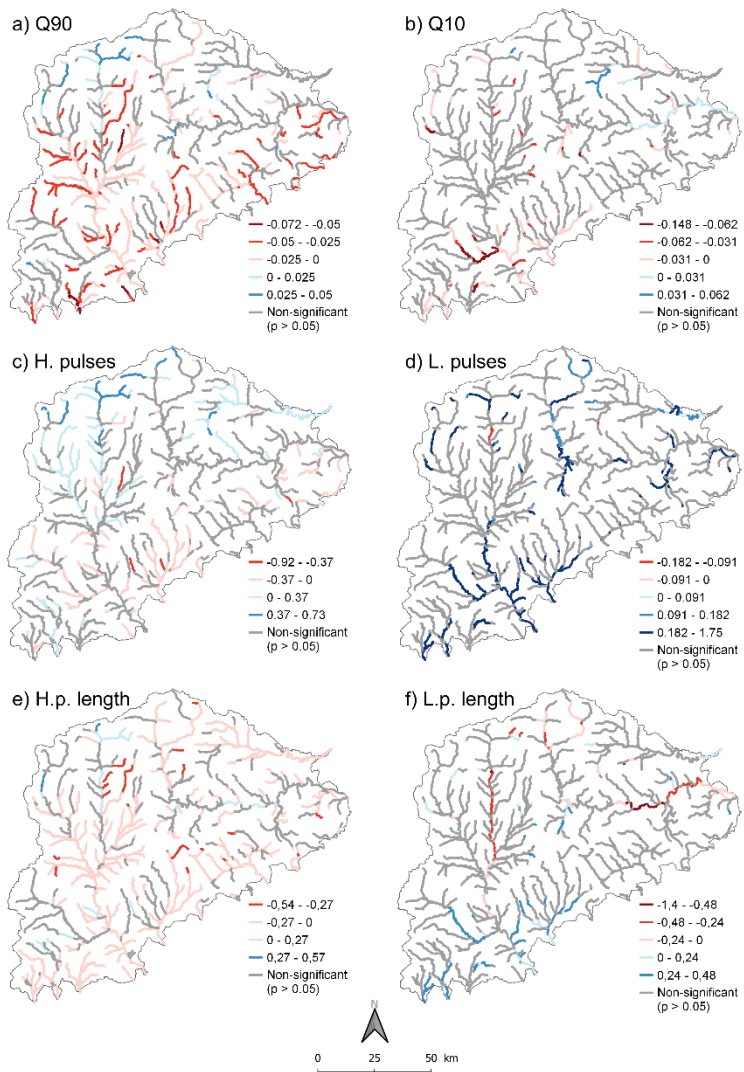

**Figure 5: Spatial distribution of Sen's slope for the hydrological indicators Q90 (a), Q10 (b), number of high and low flow pulses (c, d), and their mean duration (e, f). The Sen's slope for Q90 and Q10 is standardized, while the units for the other indicators are**

**number of events/year (c, d) and days/year (e, f).**



Despite the general decrease in high flows in the Llobregat basin, the annual maximum daily flow shows an overall positive trend. In other basins, trends vary, with some displaying mixed patterns (Besòs, Foix) and others primarily characterized by non-significant trends or limited negative trends.

Zero-flow indicators complement and expand the information that is not captured with low-flow indicators, as most of the
river network shows some degree of intermittency. The number of days with zero-flow per year has notably increased, with Sen's slopes of up to 11 days/year in many segments in the Llobregat and Besòs basins (Figure 6a). There are very few segments, all tributaries of the Ter River, with a decrease of zero-flow days. Trends in the mean duration of zero-flow events show mixed results, but a slight tendency towards shorter and longer zero-flow events can be observed in the southern and northeastern regions respectively (Figure 6b). We observe a general tendency towards an earlier onset of zero-flows, in the
scale of several days and up to a week per year (Figure 6c), while more mixed results can be observed for the central point of zero-flows (Figure 6d).

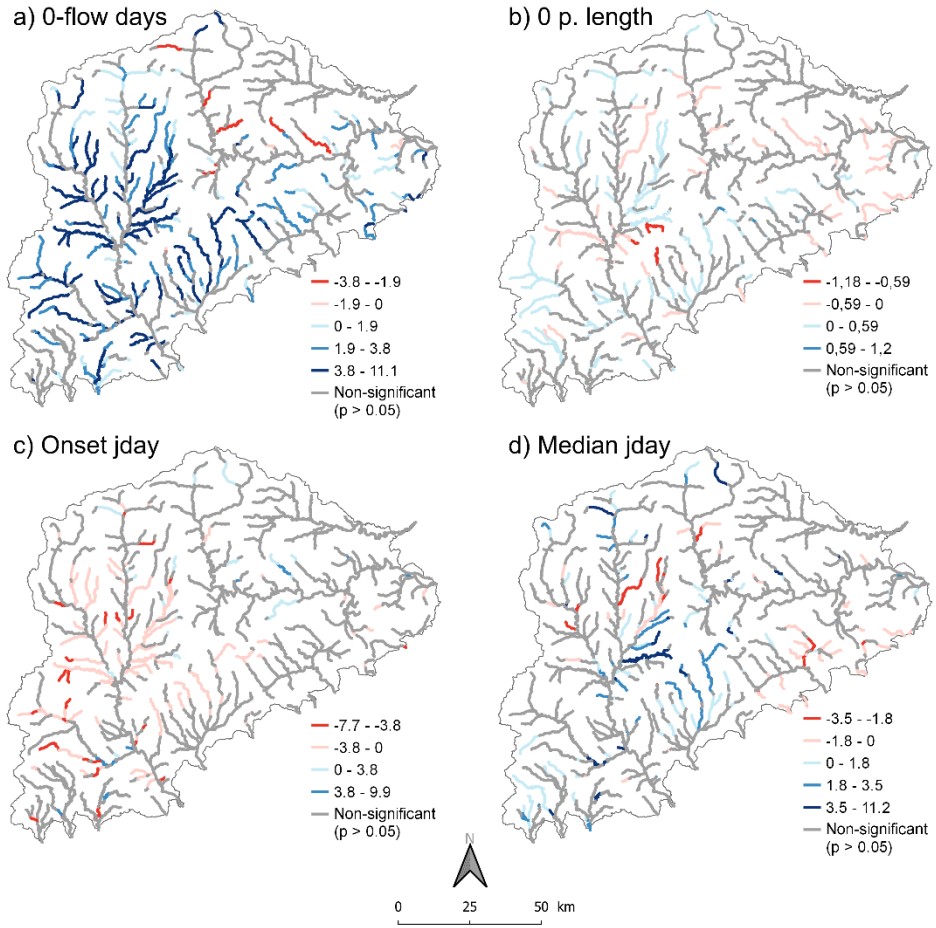

**Figure 6: Spatial distribution of Sen's slope for the hydrological indicators total annual days with zero-flow (a), mean duration of zero-flow pulses (b), julian day of first zero-flow (c), and median julian day of zero-flows (d). Sen's slope units for all indicators are**
**days/year.**



Regarding trends in hydrological indicators informing of the rate of change, there is a clear tendency towards increased rise (Figure 7a) and fall (Figure 7b) rates, except for the Besòs basin. The number of reversals tend to increase in the Pyrenees/northern regions, while they decrease in the Besòs and Foix basins, as well as some notable tributaries in the Llobregat basin (Figure 7c).

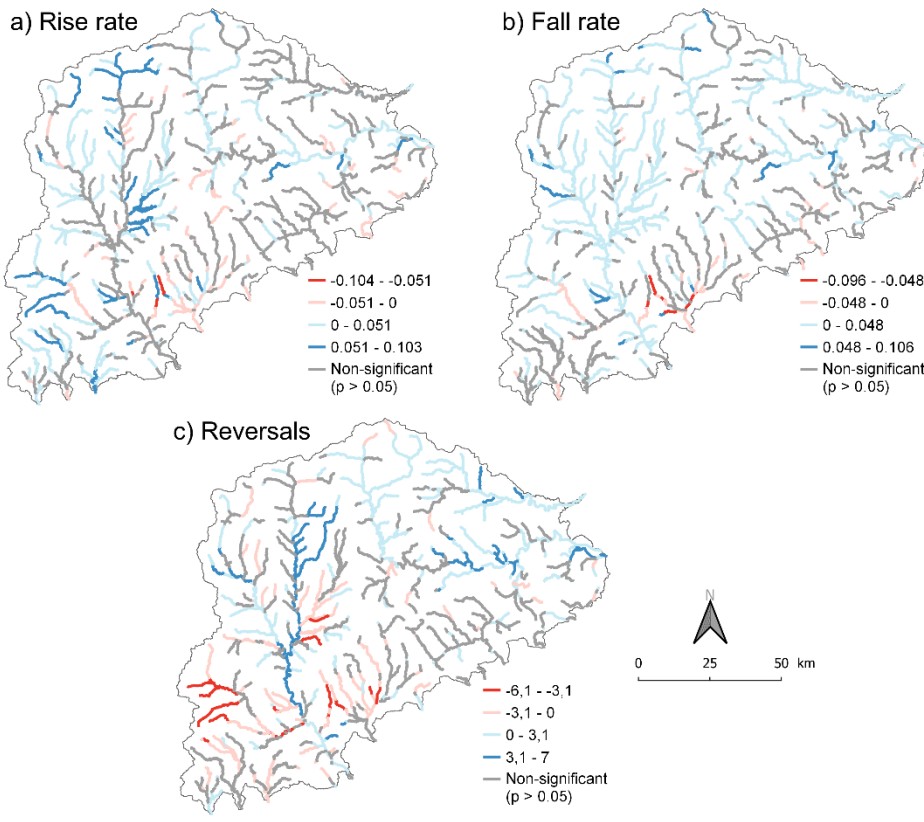


**Figure 7: Spatial distribution of Sen's slope for the hydrological indicators rise rate (a), fall rate (b) and number of flow reversals (c). The former's units are number of reversals/year for the last indicator, while the Sen's slope is standardized for the rise and fall rates indicators.**

## 4 Discussion

### 4.1 Shifting integrative hydrological modelling

Before proceeding with the discussion of the results, several considerations concerning the followed approach should be discussed. As highlighted in the introduction, the incorporation of anthropogenic activities in hydrological modelling plays a crucial role in accurately capturing watershed dynamics and supporting decision-making. We integrated anthropogenic activities in our SWAT+ model such as crop scheduling, irrigation, urban water abstractions, discharges from wastewater

treatment plants and industries, water transfers between basins, and reservoir release operations. The last one is especially



important due to the profound impact of reservoirs on natural flow regimes both regarding timing and magnitude. We used expert knowledge on how releases are managed to construct custom decision tables for each reservoir. However, actual releases depend on other factors besides the ones that can be included in decision tables, such as hydropower demand and preventive releases for flood risk mitigation. As a result, our approach approximates real management operations, much like

the other anthropogenic factors in our model, which are also based on estimations and generalizations. Another consideration on model inputs is the fact that land use change during the simulation period is not considered in our study, assessing instead only the climatic and direct anthropogenic effect on streamflow.

We introduced a novel approach to unbiasedly determine the calibration and validation periods suitable for case studies where gauging data is abundant but spatio-temporally heterogenous. We consider that, given the high number of gauging

stations in our study, randomly determining the calibration and validation periods for each station effectively captures all spatio-temporal variability in both periods. Thus, a bootstrapping method to repeatedly resample the calibration and validation periods, which is time consuming and would imply running many more iterations, is not necessary. However, the validity of this consideration could be further explored in future applications, as well as other approaches based on statistically similarities in cases where gauging data is less abundant and a random split would not be sufficient.

Hydrological indicators, such as the ones used in this research to identify trends, or techniques such as persistent homology (Musa et al., 2022), could be used to characterize hydrological behaviour or gauged data and define statistically similar calibration and validation periods. Even though we explored using such methods in our study, it would have required analyses for each individual gauging station, as well as developing a more complex methodology with uniform selecting criteria, and ultimately, we assumed that for our study the random approach would be sufficient to capture the spatio-

temporal variability.

The calibration and validation results are satisfactory overall (Table 4) and for many of the gauging stations, however, some gauging stations show poorer results. This might be due to the quality/length of the recorded data and the spatio-temporal distribution of weather stations within the draining area. Some gauging stations present periods with incongruous data, which have been removed, but some less evidently inconsistent data may remain and affect model performance. Moreover, in

stations with shorter records the calibration and validation periods might end up being too different from each other due to the random approach, and this could also affect the performance of individual gauging stations, despite their reduced weight in determining overall results. Another important factor affecting local model performance is the inaccuracy of the default SWAT+ aquifer module in accurately representing hydrogeological processes. The integration of complex hydrogeological processes and their interaction with surface water in hydrological is required to more accurately simulate watershed

dynamics (Frey et al., 2021). Future studies should prioritize the refinement of models to better capture these intricate processes, using for example the more advantageous gwflow module for SWAT+ (Bailey et al., 2020). One region with poorer performance, as it has been noted in the results section, is the headwaters of the Llobregat basin, due to it being a mountainous heterogenous area with complex karstic geology, as well as not having enough coverage of meteorological stations to properly capture the complex rainfall distribution.



The last consideration does not concern the model but rather the spatio-temporal analysis of streamflow patterns and trends. A modified Mann-Kendall test was used to account for temporal autocorrelation, but spatial autocorrelation (i.e., the effect that upstream segments have on downstream segments) was not considered in the analysis. We explored the possibility of removing spatial autocorrelation (e.g., using spectral clustering), as it could provide insights into factors influencing streamflow aside from network connectivity, but ultimately, we deemed it non-relevant for the purpose of this paper's

analysis. However, further studies could expand on removing spatial autocorrelation as complementary information to conventional analysis.

## 4.2 Patterns and trends of streamflow

The analysis of patterns and trends of streamflow for the first two decades of the 21st century in the CRBD has revealed an overall trend indicating a reduction of water resources availability, which is consistent with other Mediterranean case studies

(Clavera-Gispert et al., 2023; Masseroni et al., 2021). Three main messages can be derived from this research's results: 1) we identified a reduction in streamflow as well as an increase in intermittency; 2) streamflow flashiness has been intensified; and 3) monthly trends reveal that all seasons have not been evenly affected by recent climate change.

### 4.2.1 Streamflow reduction and increased intermittency

Significant decreasing trends over the CRBD have been identified for medium and high flows (Figure 4a & 5a), as well as

total annual streamflow (Figure 3b). There has also been a reduction in high-flow events, except in the Pyrenees/northern region (Figure 5c), as well as in their duration (Figure 5e). The regional positive trend in high-flow events is most probably due to complex spatio-temporal patterns of rainfall distribution (Lana et al., 2021) which produce high-flow events, but the rest of the indicators reveal a decreasing trend in water resources availability, and this decreasing tendency has been predicted to continue during the 21st century (Caretta et al., 2022; Marx et al., 2018). Most trends on low-flow and number of

low-flow events are not significant (Figure 5b,d), mainly because the 10th percentile which defines the low-flow threshold is zero in many river segments, but the few significant trends also reveal a decreasing tendency. Zero-flow indicators are more informative, showing an increase in number of days with zero-flow (Figure 6a) and in the percentage of river segments that dry at least once a year (Figure 3a), as well as an earlier annual onset of zero-flow (Figure 6c). This increase in flow intermittency has also been identified in other Mediterranean case studies (Llanos-Paez et al., 2023; Tramblay et al., 2021).

Precipitation in the Mediterranean region is projected to decrease over the following decades, which is expected in turn to decrease water resources availability (Dai et al., 2018; MedECC, 2020). However, in this study we identified a tendency towards flow reduction and increased intermittency while no decreasing trend in precipitation is observed for the period 2001-2022 (although there is a clear inter-annual correlation, see Figure 3a). Khorchani et al. (2021), Peña-Angulo et al. (2020), and Vicente-Serrano et al. (2022) have reported similar findings. This implies that other factors than precipitation

affect streamflow, such as temperature (and consequently evaporative demand), which in our study does in fact follow an increasing tendency and it is also projected to increase in the following decades (Ali et al., 2022; MedECC, 2020).





Land use change has also been identified as an important factor influencing streamflow (Dennedy-Frank and Gorelick, 2020), and numerous studies in the Mediterranean region have explored the impact of these land use changes on streamflow (Buendia et al., 2016; Gallart et al., 2011; Gallart & Llorens, 2004; Khorchani et al., 2021). These studies report a correlation

between afforestation in the headwaters and decreases in streamflow. Rural exodus during the second half of the 20[th] century in the CRBD resulted in the expansion of forests on abandoned cropland and pasture land (Cervera et al., 2019). During the first two decades of the 21[st] century this trend has continued, with forested area going from 46% of our study area in 2000 to 56% in 2018 (EEA, 2000, 2018). To simplify the modelling exercise, we did not account for land use changes, although we did include the effects of increased evaporation due to warming, which may be more relevant (Buendia et al., 2016).

However, the fact that we did not include direct land use changes means that we cannot attribute exact changes in streamflow to climate change, land use change or other factors.

To sum up, the reduction in streamflow during the following decades can be expected to be even more severe due to the combined effects of climate change, land use change, and rising anthropogenic demands, thus reinforcing the need for sustainable water resources management.

**4.2.2 Increased streamflow flashiness**

An increase in streamflow flashiness has also been observed. Flashiness refers to the frequency, rapidity, and magnitude of short-term changes in streamflow (daily changes in this study).

Despite the overall tendency towards a reduction in streamflow, the annual maximum daily flow has increased in the Llobregat, Besòs and Foix basins. Similar results have been found by Bouadila et al. (2020) and Varlas et al. (2023). This

trend might be due to increased precipitation extremes events (Ali et al., 2022), which are projected to keep increasing in the following decades in spite of total precipitation reduction (Zittis et al., 2021). Therefore, Mediterranean basins will suffer from both water scarcity and from increased flood risk, making the role of water management even more prominent to defend vulnerable regions.

Changes in annual maximum are not the only evidence of intensified streamflow flashiness. Trends in median rise and fall

rate indicate a generalized increase in daily streamflow changes (Figure 7a-b), and the annual number of reversals also shows a positive trend in most regions (Figure 7c). All this implies that day-to-day changes in streamflow are both more significant and more frequent, thus increasing flashiness, which can have severe impacts on ecosystem function (Pletterbauer et al., 2018).

**4.2.3 Uneven seasonal response to climate change**

Most hydrological indicators are calculated and assessed at an annual scale. However, the Mediterranean climate is characterized by marked seasonality with hot, dry summers and mild winters, with most rainfall occurring in autumn and spring. Therefore, in this study we also analysed trends for monthly median flow to characterize intra-annual dynamics and different seasonal patterns.




The most relevant tendency is that of autumn median flows (Figure 5e), where most significant trends are negative. The
Llobregat basin, of especial interest for water management, is particularly affected. Trends in winter flows also reveal a
notable decrease in the Llobregat basin, but positive trends are observed in northeastern basins (Figure 5b), which might be a
result of reduced snowfall in the Pyrenees due to increasing temperature (Döll and Schmied, 2012). Many spring and
summer trends are non-significant, but negative trends prevail (Figure 5c-d).

Although decreasing trends in streamflow are prevalent for all seasons, their extent and magnitude differ. Autumn is the
most affected season, followed by winter and summer, while spring remains almost neutral. The marked reduction in autumn
flows, which has traditionally been the wettest season in CRBD, has also been observed in Clavera-Gispert et al. (2023). The
extension of summer conditions into autumn, which were exceptionally anomalous in 2022 (Meteocat, 2023), is projected to
continue (Spinoni et al., 2018). Delayed or lack of autumn rains put significant pressure on reservoirs and aquifers, already
depleted after dry summers with higher demands, and exacerbates water scarcity.

### 4.3 Hydrological modelling in water-scarce regions and implications for water management

Process-based hydrological modelling is a useful tool to characterize and predict water resources availability within a
watershed, which then has the potential to support decision-making in the water sector, especially if management operations
and infrastructures are integrated into the modelling framework. Climatic and/or management scenarios can be implemented
and assessed to determine the best measures to implement to support sustainable and resilient societies and ecosystems.
Integrating the management factor is especially important when modelling water-scarce regions, characterized by both an
intrinsic limitation on water resources availability, which can be exacerbated by climate change, and growing demographic
pressures.

The Mediterranean has been identified as one of the most critical regions where the issue of water scarcity will be aggravated
by climate change (Cramer et al., 2018), to the point of being described as a climate emergency (Bremberg et al., 2022).
Therefore, the importance of hydrological modelling to support water resources management will continue to grow, and
modelers and managers need to communicate effectively so that hydrological modelling can be successfully integrated into
the planning and management processes.

Our approach led to successfully simulating hydrological and anthropogenic processes in water-scarce Mediterranean basins.
During model development there was collaboration with water managers from the Catalan Water Agency, the CRBD's
authority, who provided valuable first-hand expert knowledge on actual management operations to include in the SWAT+
modelling framework, as well as participated in several follow-up meetings and SWAT+ training sessions. This co-
development with water managers improves the quality of the hydrological model in simulating management operations and
helps address specific problems and concerns of water managers, while also allowing water managers to get more
familiarized with the model's capabilities and its use during planning and management. In particular, the inclusion of expert
knowledge on reservoir operations in custom decision tables allows us to simulate real reservoir releases more accurately,
and it also provides water managers with an intuitive tool to test different management scenarios and thus support decision-




making on this topic. Past applications of decision tables for reservoir release operations based on generic rules, which cannot reproduce complex reservoir-specific management guidelines (Chawanda et al., 2020; Arnold et al., 2018), thus limiting their application and reinforcing the need to integrate this expert knowledge to better simulate releases.

The good model performance achieved for the validation period corroborates our assumption that the new strategy proposed to define the calibration and validation periods randomly for each individual gauging station captures all spatio-temporal variability and it is a good approach when dealing with spatio-temporally heterogeneous basins. However, as discussed in section 4.1, this strategy could be improved by using some other criteria to replace the random selection and accounting for statistical similarity between periods.

Future developments should also focus on further improving the integration of anthropogenic pressures and processes into hydrological modelling. Moreover, as already discussed in section 4.2.1, land use changes can have a profound impact on water resources availability and distribution and their integration into the modelling framework can be of great interest, especially when considering future management scenarios. Another interesting path for future research is the development of more easily accessible and operable hydrological models, such as on-line tools and river basin digital twins which allow for

the continuous estimation of streamflow using easily available meteorological forecasts.

**5 Conclusions**

There is a pressing need to promote sustainable water management, especially in regions vulnerable to water scarcity, such as the Mediterranean, where rising anthropogenic demands and increased severity of droughts due to climate change are expected to exacerbate water scarcity in the following decades. In this context, hydrological modelling that can accurately

simulate both natural and anthropogenic processes to assess water resources availability stands as a useful tool to support efficient decision-making in such water-scarce regions. In this research, we integrated first-hand expert knowledge on management operations into the hydrological model SWAT+ and applied it in the CRBD, located in the western Mediterranean.

We used the SWAT+CRBD model to characterize and identify spatio-temporal patterns and trends of streamflow for the

period 2001-2022, to deepen our understanding of how climate change and anthropogenic dynamics impact water-scarce basins. We identified a tendency towards flow reduction and increased streambed drying throughout the study region despite not observing a decreasing trend in mean annual rainfall, which indicates that other factors such as increasing temperature and anthropogenic pressures have a profound impact on water resources availability.

The integration of first-hand expert knowledge from water managers into our modelling framework, along with the

introduction of a novel random calibration and validation approach, have resulted in notable improvements in hydrological modelling and its potential use to support decision-making in the water management sector. The spatio-temporal analysis of streamflow patterns and trends have also provided insights into the evolution of hydrological dynamics under climate change and increasing anthropogenic pressures in basins vulnerable to water scarcity.



**Code and data availability**

The modified SWAT+ source code is available from the repository https://doi.org/10.5281/zenodo.10362245. All data used and presented in this paper are available upon request.

**Author contribution**

LE: Conceptualization, Data curation, Formal analysis, Methodology, Visualization, Writing – original draft preparation, Writing – review & editing. XG: Data curation, Writing – review & editing. JS: Software, Writing – review & editing. RM:
Supervision, Writing – review & editing. AM: Data curation, Project Administration, Writing - review & editing. VA: Conceptualization, Funding acquisition, Project Administration, Supervision, Writing – review & editing.

**Competing interests**

The authors declare that they have no conflict of interest.

**Acknowledgements**

This research was funded by the Catalan Water Agency under the contract CTN20A0533, titled "Analysis of the potential urban sanitation improvements and industrial point source contaminants traceability". L. Estrada acknowledges funding from the Secretariat of Universities and Research of Generalitat de Catalunya and the European Social Fund for her FI fellowship (2023 FI-2 00168). Authors acknowledge the support from the Economy and Knowledge Department of the Catalan Government through Consolidated Research Groups (ICRA-ENV 2021 SGR 01282), as well as from the CERCA program.

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
