# Peer review of "Spatio-temporal patterns and trends of streamflow in water-scarce Mediterranean basins"

_EGUsphere, 2023_

## Author Response (AR1)

**Reviewer 1**

The manuscript "Spatio-temporal patterns and trends of streamflow in water-scarce Mediterranean basins" by Laia Estrada et al. proposes an intricate modelling exercise on a set of drainage basins of the Catalan River Basin district, with the main objectives of providing a tool useful for water management and assessing the spatio-temporal patterns and trends of stream flow during the first two decades of the 21st century.

The purposes and the extent of the exercise as well as the contribution of the stakeholders and the innovative design of the model calibration-validation are the main strengths of the manuscript. Nevertheless, there are severe methodological inadequacies that not only put into question model results but also would provide inadequate examples on how this kind of exercises may be correctly done.

Response: Thank you, we found your feedback very helpful! We have done our best to address all your comments, especially the methodological issue arising from excluding the land use changes during the simulated period, to improve this manuscript.

Working hypotheses:

In the Discussion section (lines 397 and subsequent), the authors quote several publications that "report a correlation between afforestation in the headwaters and decreases in streamflow" and indicate that "during the first two decades of the 21st century this trend has continued, with forested area going from 46% of our study area in 2000 to 56% in 2018".

Disregarding these clear alerts, the authors "did not account for land use changes, although we did include the effects of increased evaporation due to warming, which may be more relevant (Buendia et al., 2016)."

Response: Thank you for your comment. We agree that we did not properly justify the exclusion of the effect of land use change on streamflow, so we performed the analysis of temporal trends of the model residuals as recommended. This analysis is presented and discussed in section 1 of the Supplement. We do not find clear evidence that the increase in forest area or density are a major factor influencing flow in our study, as the number of positive and negative trends in the model's residuals are equivalent and cannot be interpreted as a single factor not included in the model being responsible for such trends, despite most of them being significant. Therefore, our assumption that climate variability is the main driver of change in streamflow and that land use change can be omitted from the modelling exercise without compromising results is not incorrect.

But Buendia et al (2016), in the Final Remarks section state: "Overall, results have indicated that increased forest areas are the major driver of reduced streamflows and the magnitude of peak floods."

Response: The sentence cited as to being part of the Final Remarks in Buendia et al. (2016) is not found anywhere in the paper. In fact, although important, increasing forested area was not found to be the major driver (Figure 9). Nonetheless, we have rewritten this section so it is better formulated.

Lines 435-441:

"However, under the assumption that climate variability and not afforestation is the main driver of streamflow reduction (Buendia et al., 2016), we did not account for these land use changes in the model, and we used the forested area in 2018 for the whole simulation period. The analysis of trends in model residuals (see section 1 of the Supplement) does not evidence the presence of a factor other than the ones already included in the model affecting the hydrological response, and thus justifies the exclusion of land use changes in our study. However, it must be noted that despite not accounting for land use changes per se, we do account for the increase in evapotranspiration due to increased temperatures."

The working hypothesis that the increase of forest cover can be omitted as possible cause of temporal trends of streamflow should be stated in the methods section, and the reason claimed by the authors for this omission is untrue.

Response: We agree that we should include our hypothesis in the methods section.

Lines 168-171:

"It should be noted that only a static rather than dynamic land use map is considered in this study, and thus we are omitting the effect that changes in land use during the simulation period may have on streamflow, under the assumption that climate and not land use change is the main driver of the hydrological response. To verify this hypothesis, we performed an analysis of the trends in the model residuals (section 1 of the Supplement)."

Given the results of several previous works, it is likely that land cover change is a more relevant driver of recent hydrological changes in this area than climate warming. Both the overall and spatial flow trends simulated by the model become highly doubtful and are not compared with actual ones.

Response: While we agree that land cover change can have an impact on hydrological changes, we disagree that is a more relevant driver than climate change in this area, as exemplified in the works cited. However, despite the fact that the trends were computed with calibrated streamflow and thus an argument can be made that they are indeed comparable to observed trends, we agree that also comparing them with the observed trends at the gauging stations could be interesting. We added this comparison in section 3 of the Supplement, as well as included the following paragraph in the 'Material and methods' section.

Lines 250-258:

"The advantage of using simulated streamflow rather than observed is working with 999 values for each indicator instead of only 50 (gauging stations), which allows us to better observe spatial patterns. Moreover, some gauging stations present gaps for the period 2001-2022, so simulated streamflow also provides a complete temporal series. Nevertheless, we compared the trends in indicators calculated with observed streamflow with the simulated trends at four gauging stations (see section 3 of the Supplement). We found significant observed trends for 20-29 hydrological indicators (50-72.5% of all indicators), and significant simulated trends for 11-23 hydrological indicators (27.5-57.5%). Most of significant simulated trends (66.7-73.3%) are also significant using the observed flow, and the majority of those (82-100%) are in the same direction (i.e. positive or negative trend). Therefore, while we do not capture all the observed trends with the model, the trends that we do capture are comparable to the observed trends."

Analysis of modelling results:

Contrary to its recurring attribution as a 'physically based model', SWAT is an empirical model without a sound physical basis. The core of SWAT is the Curve Number Model that is undoubtedly an empirical model.

This is not just a rhetorical question but is relevant to the interpretation of the modelling results. In a physically based model there might be some hope that the internal model variables (stores and fluxes) are acceptable if the simulated discharge is so (but Anderton et al., 2002). However, when a conceptual or empirical model is calibrated using streamflow data, "Model performances measure the correctness of estimates of hydrological variables generated by the model and not the structural adequacy of the model vis-à-vis the processes being modelled" (Klemes, 1986). In other words, the model not necessarily gives the "good answers for the good reasons" (Grayson et al., 1992; Beven, 2002; Kirchner, 2006), so model fluxes not directly used for calibration are highly suspect of being model artefacts.

Response: Thank you for your insight. However, we disagree that SWAT is "an empirical model without a sound physical basis". While it is true that some processes might have an empirical basis (such as the runoff/infiltration with the curve number model), SWAT+ is a very complex model that simulates and parameterizes a large number of biophysical processes. The soundness and usefulness of the SWAT and SWAT+ models have been proved in many studies worldwide (see many referenced works throughout the paper).

Furthermore, the uncertainties associated to the model simulations (at least those used for calibration) must be analysed to provide the users with estimates of the risks in decision making (Grayson et al., 1992; Beven and Binley, 1992; Beven 2006; Herrera et al.,2022...).

Response: Thank you for your comment. We have included the quantification of the uncertainty associated to simulated streamflow in section 2 of the Supplement.

We also separated Fig. 3 into two figures, and for Fig. 4 (Annual Flow) we added the distribution of non-standardized Sen's slope and of the standard deviation.

"We also assessed the 95PPU uncertainty bands and their metrics P-factor and R-factor (Abbaspour et al., 2015, 2018) for representative gauging stations of each main basin (see section 2 of the Supplement)."

"The comparisons between observed and simulated daily streamflow for representative gauging stations of the main rivers of the CRBD demonstrate good model performance (Fig. 2, see section 2 of the Supplement to visualize the uncertainty represented by 95PPU bands)."

"We don't observe a specific spatial pattern on the distribution of the Sen's slope standard deviation for total annual flow (Fig. 4c), except for the few significant trends in the Tordera basin, where the standard deviation is generally high, so overall we can conclude that the uncertainty in Sen's slopes for all CRBD is similar."

Figure 4:

[Figure]

**Figure 4: Spatial distribution of Sen's slope (a, units hm3/year), standardized Sen's slope (b) and standard deviation (c) for the hydrological indicator total annual flow.**

Finally, In a sub-section section of the 'Materials and Methods' section named 'Data analysis', the authors included the calculation of many hydrological indicators, but contrarily to the title of the sub-section, this analysis was made (if I am not in error) not on the original 'data' but on internal (not calibrated) model results. Therefore there is no assessment on how these indicators represent the ones of the actual hydrological regimes.

Response: The analysis was indeed made with model results, albeit they are calibrated, and thus it can be argued that the indicators analysed are comparable to those computed with observations. Moreover, using calibrated streamflow instead of observed allows us to obtain a more comprehensive spatio-temporal analysis than the one the observed record can provide, as it presents gaps in space and time. However, following a previous comment, we have already added the comparison between observed and simulated trends for some gauging stations in section 3 of the Supplement. We have also changed the section title to 'Trend analysis' to avoid confusion.

Overall manuscript assessment.

In spite of the valuable strengths stated above, the modelling exercise is based on the inadequate working hypothesis that warming is the main driver of hydrological trends in this area and manages several principal and internal model outputs as actual data without any assessment of the uncertainty associated with these simulations.

Response: Thank you for your comment. As mentioned in other responses, the hypothesis that climate change is the main driver or hydrological trends in this area is not inadequate, as discussed in section 1 of the Supplement.

Concerning the uncertainty, we have quantified and shown the uncertainty of simulated streamflow, as well as for one of the main indicators Annual Flow (Fig. 4 and Supplement section 2).

Recommendations.

Both the importance of the objectives and the magnitude of the modelling exercise deserve finding some feasible way to improve the soundness of the project.

Response: Thank you for your comment. We hope that with your recommendations we managed to improve this manuscript.

The fact that the encroachment of forest cover in the studied catchments is a likely or very likely driver of the hydrological response involves a difficulty for the modelling exercise but an opportunity for water management. Indeed, if climate were the main driver of the hydrological response, management strategies for adaptation to the climate change would be limited. Conversely, if forest cover is the main driver, it can be managed to reduce the 'green water' consumption and increase the 'blue water' delivery (Falkenmark, 2000) as a climate change adaptation strategy.

Using SWAT for simulating the hydrological response to forest cover change is a cumbersome and risky task, taking into account the poor or very intricate examples available (Haas et al., 2022; Karki et al., 2023).

Response: While we have verified that climate is our main driver of the hydrological response, we do not disregard the effect of land use change and management, evidenced by its inclusion in the discussion of our article. In fact, the SWAT+CRBD model was used in Garcia et al. (2024) to assess the impact of forest management on water resources availability. We have included the following into the discussion:

Lines 361-369:

"Another consideration on model inputs is the fact that land use change during the simulation period is not considered in our study, due to having determined that land use change and in particular afforestation is not a main driver of hydrological response for the scope of this study (section 1 of the Supplement). However, it can still be an important

factor at the local scale, and its consideration represents an opportunity for future management practices. Forest cover can be managed to reduce "green water" (i.e., water stored in the soil and vegetation and that is then consumed) and turn it into "blue water" (i.e., runoff), increasing water availability in potential areas suffering from water scarcity. Garcia et al. (2024) used the SWAT+ CRBD model to assess the effect of forest thinning on water yield, and results highlighted the potential of forest management to enhance "blue water" availability."

But the flow simulations made may be used to test the null hypothesis that the climatic forcing is sufficient to explain the observed flow records, analysing whether there are time increasing model residuals that could be attributed to the role of increasing forest cover extent or density. This exercise may be made in most of the gauging stations used, providing a map of the hydrological changes attributable to the encroachment of forest cover. The statistical significance of trends should be made following the recommendations issued by the IPCC (Mastrandrea et al., 2010).

Response: Thank you for your recommendation. As mentioned in previous comments, we have performed the trend analysis of the model residuals to test whether the assumption that climatic forcing is enough to explain the observed flows (section 1 of the Supplement). We have assessed that climate variability is the main driver of change in streamflow in our study, and therefore land use change can be omitted from the modelling exercise without compromising results.

Unfortunately, the hydrological indicators analysed in the manuscript may be obtained for the flow records at the gauging stations, but any comparison with the simulated ones is expected to give inconsistent results because it is not possible to determine if the differences are attributable to modelling errors or to the role of the hydrological role of forest encroachment.

Finally, the maps of figures 4 to 7 should be discarded because these results are highly suspect of being modelling artefacts because do not take into account the role of forest cover change and these are internal model outputs not calibrated and of unknown uncertainty.

Response: As we have determined that the increased forested area is not a main driver of hydrological change in our study, the hydrological indicators obtained from calibrated streamflow are adequate for the analysis of trends and patterns. However, a comparison with observed trends has also been included, as well as the quantification of the uncertainty. Thus, we do not believe Figs. 4 to 7 (now Figs. 5 to 8) should be discarded.

**Reviewer 2**

The spatio-temporal analysis of streamflow patterns and trends over the last 20 years in Spanish Catalonia proposed in the article is, in my opinion, useful and interesting.

A large number of measurements are included in the article, and a major modeling effort is made to generate flow series that are continuous in space and time from data that are mostly discontinuous (which is always more or less the case everywhere). The analysis of patterns and trends involves numerous indicators of different flow characteristics (magnitude, duration, frequency). This makes for interesting and original results.

Response: Thank you! Your feedback has been greatly valuable in order to improve this manuscript. We have carefully read your comments and done our best to properly address them.

The authors state 3 objectives for their study:

1. Develop a useful modeling tool for water management
2. Propose a new calibration strategy that overcomes conventional approaches
3. Characterize spatio-temporal patterns and trends of streamflow.

In my opinion, the demonstration made in the article for the first 2 objectives is not completely satisfactory.

Response: Thank you for your comment. We agree with the reviewer's opinion that we should prioritise the third objective, as the other two are secondary objectives/features of the methodology. We have reestructured the Introduction to clarify that.

**On the first objective**, the authors mention in section 2 "Co-development with end-users".

The "end-users" are not clearly defined (what type of structure do they belong to? how many are there? how were they chosen? did some refuse to participate? what was their level of appropriation of hydrological sciences and modeling?)

Response: We have clarified who the intended "end-users" of our model are and their role in model development.

Lines 137-144:

"In order to familiarize end-users (i.e., water managers from the Catalan Water Agency, the governing body of the CRBD) with the hydrological model and promote its use as a tool to support decision making in the CRBD, we have actively involved them in the development of the SWAT+ model from conceptualization to application. The main role of end-users in model development has been to procure data, including weather, streamflow, point source discharges, and, more notably, expert knowledge on actual management practices. This expert knowledge includes real reservoir release operations, which were adapted into custom-built decision tables, as well as irrigation practices, interbasin water transfers, and urban abstractions. The integration of management practices results in a more accurate hydrological model with the potential use of testing different management scenarios, and thus support better informed decision making."

The authors indicate that they aim to help end-users understand how the model works, through training sessions. No details are given on the number of these sessions, their content or the end-users' prerequisites. No feedback is offered or analyzed on this appropriation phase (were there any evaluations following the training sessions? how did the end-users progress? what mastery levels were reached?).

Response: Unfortunately, some of the meetings to discuss model development and the SWAT+ training sessions were informal meetings, and so we did not keep a full record. Also, there was no formal evaluation of the training sessions, as they were solely treated as workshops where the SWAT+ model was built so the end-users could familiarize themselves with the model's interface and outputs.

The authors insist on "the inclusion of valuable expert knowledge on actual management practices". Is this to be understood as co-development of the model? Or rather as consultation to parameterize the water use rules of the reservoirs (in the same way as soil experts would have been consulted to parameterize soil properties in the model)?

In my opinion, the full description of the methods and the retrospective analysis are insufficient for this issue to be presented as an objective of the article. It would seem more appropriate to treat this point as a step in the parametrization of the model, based on expert data from the field. The authors' view of their collaboration with stakeholders could be discussed in the article, but as it stands, I don't consider that there is a clear demonstration of co-construction (what would have been the results of the modelling without this consultation on water use rules?).

Response: As per a previous comment, we have already restructured the Introduction to give greater weight to the main objective of this article. However, we do consider the involvement of end-users/water managers as co-development rather than simply consultation, as they have provided numerous data and participated in discussions on how to better incorporate it into the model. In fact, one of the co-authors is a representative of the end-users involved in the model development.

With regard to the **second objective**, the proposed calibration/validation technique is interesting, but raises a number of unresolved questions.

In my opinion, the authors give this technique an exaggerated benefit in relation to the results shown in the article. Does this technique really provide better results than a traditional calibration/validation technique? It's quite possible, but it's not demonstrated in the article. In my opinion, it would require an article of its own to demonstrate this. This technique could simply be presented in the "materials and methods" section. Its positive aspects and limitations could be discussed. But positioning it as an objective of

the article seems too strong, as do the claims that it best captures the spatio-temporal variability of hydrological processes in the study area.

Response: Thank you for this comment. We agree that the benefits of the calibration/validation strategy presented here are not quantified. In fact, for a time we considered working on a separate article where different calibration/validation strategies, including the one presented in this article, were tested. Unfortunately, the main researcher working on this exercise left the project before it could be completed. Nevertheless, we do believe our strategy has a sound basis and represents an improvement to conventional techniques. As per previous comments, we have already restructured the objectives, focusing on what was originally the third one and mentioning the other two more as secondary objectives. However, we do believe it is appropriate to introduce this topic in the Introduction rather than exclusively presenting it in the "Materials and methods" section.

Another point concerns the fact that only one land use is considered over the entire period, even though it may have varied, as the authors indicate. Would it have been more appropriate to calibrate the model on the flows of the period when this land use was in place? rather than calibrating on random periods?

Response: Thank you for your comment. Only calibrating the model using the flows from when the final land use was in place would result in a significant reduction of available data, which would in turn compromise the calibration/validation strategy. However, as a result of the first reviewer' comments, we have further addressed the implications of not taking the land use change into account, as well as justified our decision in section 1 of the Supplement.

Lines 168-171:

"It should be noted that only a static rather than dynamic land use map is considered in this study, and thus we are omitting the effect that changes in land use during the simulation period may have on streamflow, under the assumption that climate and not land use change is the main driver of the hydrological response. To verify this hypothesis, we performed an analysis of the trends in the model residuals (section 1 of the Supplement)."

Lines 361-369:

"Another consideration on model inputs is the fact that land use change during the simulation period is not considered in our study, due to having determined that land use change and in particular afforestation is not a main driver of hydrological response for the scope of this study (section 1 of the Supplement). However, it can still be an important factor at the local scale, and its consideration represents an opportunity for future management practices. Forest cover can be managed to reduce "green water" (i.e., water stored in the soil and vegetation and that is then consumed) and turn it into "blue water" (i.e., runoff), increasing water availability in potential areas suffering from water scarcity. Garcia et al. (2024) used the SWAT+ CRBD model to assess the effect of forest thinning

on water yield, and results highlighted the potential of forest management to enhance "blue water" availability."

"However, under the assumption that climate variability and not afforestation is the main driver of streamflow reduction (Buendia et al., 2016), we did not account for these land use changes in the model, and we used the forested area in 2018 for the whole simulation period. The analysis of trends in model residuals (see section 1 of the Supplement) does not evidence the presence of a factor other than the ones already included in the model affecting the hydrological response, and thus justifies the exclusion of land use changes in our study. However, it must be noted that despite not accounting for land use changes per se, we do account for the increase in evapotranspiration due to increased temperatures."

On **the third objective**, the trend analyses are really interesting and raise several questions as to their interpretation.

Trends and patterns are based on model simulations. Calibration/validation performance is uneven between periods and between basins. I think it would be useful to associate a level of confidence with the indicators produced, depending on the quality of the modeling. This would allow us to temper the conclusions regarding patterns and trends.

Response: Thank you for your comment. We have included the quantification of the uncertainty associated to simulated streamflow in section 2 of the Supplement.

We also separated Fig. 3 into two figures, and for Fig. 4 (Annual Flow) we added the distribution of non-standardized Sen's slope and of the standard deviation.

"We also assessed the 95PPU uncertainty bands and their metrics P-factor and R-factor (Abbaspour et al., 2015, 2018) for representative gauging stations of each main basin (see section 2 of the Supplement)."

"The comparisons between observed and simulated daily streamflow for representative gauging stations of the main rivers of the CRBD demonstrate good model performance (Fig. 2, see section 2 of the Supplement to visualize the uncertainty represented by 95PPU bands)."

"We don't observe a specific spatial pattern on the distribution of the Sen's slope standard deviation for total annual flow (Fig. 4c), except for the few significant trends in the Tordera basin, where the standard deviation is generally high, so overall we can conclude that the uncertainty in Sen's slopes for all CRBD is similar."

[Figure]

**Figure 4: Spatial distribution of Sen's slope (a, units hm3/year), standardized Sen's slope (b) and standard deviation (c) for the hydrological indicator total annual flow.**

Several causes are cited for interpreting flow trends: precipitation, rising temperatures (which should lead to an increase in evapotranspiration) and changes in land use.

Be careful, however, as the evolution of these causes in relation to flow changes is not quantified: l.392-393 the authors mention an absence of trend in annual rainfall, which is not quantified by a test. In addition, there may be trends in rainfall at other time steps and key periods in the year that influence river intermittency.

Response: Thank you for pointing this out. We agree that we should evaluate and quantify the trends in annual rainfall (as well as temperature) instead of only visually interpreting them, so we have added Table 5 with the quantification and significance of these trends. We have also tested the correlation mentioned between annual % of dry river segments and annual precipitation.

Lines 285-288:

"Figure 3 shows the evolution of the percentage of river segments that dry at least once a year for the period 2001-2022. We observe a drying tendency in the CRBD, which can be positively correlated to an increase in mean annual temperature (Table 5). However, while individual annual percentages negatively correlate with mean annual rainfall (Pearson's r = -0.52, p < 0.05), there is no significant decreasing trend in the latter (Table 5)."

Table 5:

**Table 5. Analysis of trends for the annual percentage of dry river segments, mean annual temperature and mean annual precipitation. Significant trends are marked in bold (p-value < 0.05). LR: Linear Regression; MK: Mann-Kendall.**

|  | LR slope | LR p-value | $R^2$ | Sen's slope | MK p-value |
|---|---|---|---|---|---|
| Percentage of dry river segments | 0.40 | **6.4E-03** | 0.32 | 0.44 | **1.9E-10** |
| Mean annual temperature | 0.05 | **2.5E-03** | 0.37 | 0.04 | **2.6E-09** |
| Annual precipitation | -0.68 | 0.90 | 8.4E-04 | -2.50 | 0.06 |

l.407-410: this summary is very probably true, but the article does not deal with forecasting future flows.

Response: We have rewritten this sentence to be more consistent with the scope of this article.

Lines 442-444:

"To sum up, the reduction in streamflow observed during the last twenty years in this study allows us to infer that this tendency will continue in the following decades due to the combined effects of climate change, land use change, and rising anthropogenic demands, thus reinforcing the need for sustainable water resources management."

In **conclusion**, it seems to me that the objectives of the article should be reformulated to focus on the 3rd objective. The other two are, in my opinion, features of the methodology and should be presented and discussed as such.

Response: Following the reviewer's comments, we have addressed the issue of the formulation of the article's objectives. Thank you very much for your insightful feedback which has allowed us to improve this manuscript.

---

## Referee Report (RR1)

General analysis: First of all, I want to apologize because the Buendia et al. (2016) paper that states "Overall, results have indicated that increased forest areas are the major driver of reduced streamflows and the magnitude of peak floods" is not the Buendia et al. (2016) paper quoted by the authors but another one (http://dx.doi.org/10.1016/j.scitotenv.2015.07.005) published two months later studying another basin. Indeed, the paper quoted by the authors found that for the embracing studied catchment (Talarn), somewhat less than 50% of the runoff reduction (37%) could be attributed to forest cover encroachment, but these authors state that "Neglecting re-vegetation could lead to erroneous projections resulting in an underestimation of the runoff future trends; thus, evolution of forested cover should not be ignored when designing land and river basin management plans at the light of global change scenarios". Therefore, this paper should not be fairly cited as a reason to omit the role of land cover change in streamflow temporal trend studies.

I want to acknowledge the effort made by the authors to follow my recommendations. The new information provided is noteworthy but not easy to understand, so I am trying to analyse it and to provide updated recommendations to the authors.

Trends: The data shown in the table S1 are really striking as they point to relevant internal inconsistencies in the model results. First, the fact that 70% of the gauging stations show positive or negative significant residual trends, with a coefficient of variation of 822%, demonstrates a high uncertainty of the model results. Second, the spatial distribution of these trends in figure S1 shows disordered patterns; some successive gauging stations without any significant tributary between them show opposite trends, such as Castellbell and Abrera stations on the Llobregat River and the two Les Masies de Roda stations on the Ter River. Third, some of the plots in figure S2 show very asymmetric abnormal shapes, either positive (Balsareny, Fogars de la Selva (Pont Eiffel), Sallent) or negative (Guixers, Sant Feliu de Buixalleu).

Negative trends of the residuals may be attributed to the role of increased forest cover in the area, a very likely behaviour already demonstrated by previous works, but here it is more difficult to attribute positive trends to hydrological reasons. I wonder whether the calibration-validation strategy used (randomly selected for each station independently) may lead to different sensitivities of the model to climate forcing at each station in such a varied climate and induce this scatter. As Buendia et al (2016) stated "precipitation appears to follow a generalised decreasing trend, although the significance of these results depended strongly on the time period considered". In fact, the authors do not provide any evidence of the validity of their consideration (line 371): "We consider that, given the high number of gauging stations in our study, randomly determining the calibration and validation periods for each station effectively captures all spatio-temporal variability in both periods. Thus, a bootstrapping method to repeatedly resample the calibration and validation periods, which is time consuming and would imply running many more iterations, is not necessary".

SWAT+: This is not a physically-based model even if it can provide with good results. This qualification of this model contributes to the degradation and loss of usefulness of terminology and concepts. The addition of complementary processes does not modify its

essentially empirical character. There is an agreed methodological caveat that just because a model gives good results does not imply that it is for the good reasons (in particular, structure, internal stores and flows). In fact, the results shown by the authors in table S1 may provide a corollary of this principle: despite acceptable flow calibration/validation tests, residual discharge trends show chaotic values difficult to attribute to hydrological reasons.

Recommendations:

In general terms, the manuscript describes the modelling exercise, shows its results, makes some comparisons with observed data and claims the success of the exercise as it "led to successfully simulating hydrological and anthropogenic processes in water-scarce Mediterranean basins" and "resulted in notable improvements in hydrological modelling and its potential use to support decision-making in the water management sector" without contributing no evidence of these successes and improvements.

The authors should not claim good modelling results beyond acceptable tests of efficiency and uncertainty, but should be much more analytical by discussing their strengths and weaknesses and suggesting ways to remedy the latter.

- The authors cannot justify the reason for the omission of the role of land cover in the hydrological changes of this area on the basis of any published work, nor justify the validity of this omission on the modelling results which are largely inconsistent in this respect.

- Following Bieger et al (2017), the authors can claim that SWAT is "one of the most widely used hydrologic models in the world" but cannot claim that it is a physically-based model.

The model parameters were optimized to obtain the best simulation of discharges, but not the various 'hydrological indicators' extensively exposed in the manuscript. Therefore, due to the equifinality problem (Beven, 2006; Kirchner, 2006), various sets of model parameter may give discharge efficiencies very similar to the best one, but may give quite divergent values of these 'hydrological indicators'.

Consequently, model-simulated 'hydrological indicators' face to two severe uncertainties: the role of land cover changes and the issue of model equifinality. It is not possible to determine whether the differences between modelled and observed trends of these indicators are due to the role of land cover change or modelling equifinality effects. Therefore, the statement in line 76 of the Supplement regarding trend analyses is not acceptable: "Moreover, the fact that streamflow was first calibrated ensures overall the validity of the analysis".

- Throughout the manuscript, the trends are shown as "Sen slopes", but the units are not always shown, especially for the time variable, so the value of the rate of change is not clear if it is per day, month or year.

Removing the stronger influence of river segments with higher streamflow in the analysis of temporal trends does not seem to me a sound option when the main objective of the study concerns water resources. On the other hand, the comparison between slopes and mean flows (fractions of runoff gained or loss at annual intervals) are convenient to evaluate their importance in terms of water resources.

- The hydrographs shown in figures 2 and S3 to S8 are very difficult to understand because the diverse plots cover each other. A logarithmic scale of the discharge axis might help to better visualize the plots.

- The units for the variables $x$ and $y$ in the equation enclosed in Figure 3 are not stated. This graph seems to mix observed and simulated results, which should be explained.

- The analysis of trends in model residuals in section 1 of the Supplement does not justify the validity of excluding land-cover changes in the study, but demonstrates the difficulty of the modelling exercise to provide reliable estimates of the trends.

- The units of the slopes shown in table S1 are not shown. Both this table and figures S1 and S2 demonstrate the very inconsistent results of the model exercise with respect to these trends. These results must be further discussed and the sentence "we can reasonably assume that land use changes in our study are not a main driver influencing streamflow" should be deleted.

- The Y-axes in figures S9 and S10 are not appropriate because variables of diverse ranks are shown together. The ratio of slope to mean value (%) could be better as Y-axe units, as this could increase the visibility of low values and allow direct comparison between gauging stations because the axes could be equal.

---

## Referee Report (RR2)

The authors have taken on the comments and suggestions made in the first review concerning the reformulation of objectives and the definition of end-user input. The supplements are interesting and useful.

In my opinion, the manuscript is acceptable in this 2nd version.

Just one last comment:

The residue study in supplement 1 is a good idea. Trends are presented basin by basin (which is interesting). But these trends are only compared with the percentage increase in forest area over the entire study area. It would have been more convincing to show the variation in forest area (and/or urbanized area) in each basin beside the residual trends.

---

## Referee Report (RR3)

As I pointed out in the previous review, the authors have taken good account of the suggestions made. The exchanges with reviewer 1 are rich. I share many of Reviewer 1's opinions, while at the same time finding the proposed exercise of reconstructing hydrological chronicles to analyze trends interesting.
To be consistent with all the discussions between reviewers and authors, I suggest a few small adjustments to the text:

L380-383
Replace
*We consider that, given the high number of gauging stations in our study, randomly determining the calibration and validation periods for each station **effectively captures all spatio-temporal variability** in both periods.*

with
*We consider that, given the high number of gauging 380 stations in our study, randomly determining the calibration and validation periods for each station **efficiently captures spatio-temporal variability** in both periods.*

L492-494
*Process-based hydrological modelling is a useful tool to characterize **and predict** water resources availability within a watershed, which then has the potential to support decision-making in the water sector, especially if management operations and infrastructures are integrated into the modelling framework*

Remove "and predict" as the article does not deal with forecasting

L494-495
The article doesn't suggest an optimization method, so I suggest replacing
*Climatic and/or management scenarios can be implemented and **assessed to determine the best measures** to implement to support sustainable and resilient societies and ecosystems.*

with
*Climatic and/or management scenarios can be implemented and assessed to **project** measures to implement to support sustainable and resilient societies and ecosystems.*

L516-518
Replace
*The good model performance achieved for the validation period corroborates our assumption that the new strategy proposed to define the calibration and validation periods randomly for each individual gauging station **captures all** spatio-temporal variability and it is a good approach when dealing with spatio-temporally heterogeneous basins.*

with
*The good model performance achieved for the validation period corroborates our assumption that the new strategy proposed to define the calibration and validation periods randomly for*

*each individual gauging station **efficiently captures** spatio-temporal variability and it is a good approach when dealing with spatio-temporally heterogeneous basins.*

L544-546

I suggest to remove "**highlighting the potential use of this model to support decision-making in the water management sector**". It's not just because you've integrated information from experts that the model is more useful or easier to access for those experts

Last line

"without the need for complex analysis", it's a bit obscure, we don't understand what kind of complex analysis you're talking about. I suggest you withdraw it.

---

## Author Response (AR2)

**Reviewer 1**

General analysis: First of all, I want to apologize because the Buendia et al. (2016) paper that states "Overall, results have indicated that increased forest areas are the major driver of reduced streamflows and the magnitude of peak floods" is not the Buendia et al. (2016) paper quoted by the authors but another one (http://dx.doi.org/10.1016/j.scitotenv.2015.07.005) published two months later studying another basin. Indeed, the paper quoted by the authors found that for the embracing studied catchment (Talarn), somewhat less than 50% of the runoff reduction (37%) could be attributed to forest cover encroachment, but these authors state that "Neglectig re-vegetation could lead to erroneous projections resulting in an underestimation of the runoff future trends; thus, evolution of forested cover should not be ignored when designing land and river basin management plans at the light of global change scenarios". Therefore, this paper should not be fairly cited as a reason to omit the role of land cover change in streamflow temporal trend studies.

Response: Thank you again for all your feedback and recommendations to improve our manuscript. We cited Buendia et al. (2016) as an example of study where land use changes, while important, are not the main driver of runoff reduction. Indeed, for the Talarn catchment 37% of runoff reduction is attributed to forest cover encroachment, and while this is a major impact and omitting this land use changes for this particular catchment could lead to erroneous results, the main driver is still climate variability. Moreover, runoff reduction attributed to forest encroachment in the other two basins in Buendia et al. (2016) is 6% and 16% respectively, showcasing how widely the impact of land use changes can vary from basin to basin.

We have reformulated this section so it is more clear that we are not omitting land use changes in our study based on Buendia et al. (2016). We are omitting them based on the analysis of trends in model residuals, as recommended in the first revision, where it is shown that there are no clear trends that could be attributed to their omission, and therefore we can proceed with our analysis. However, we are not negating the importance of land use changes and their impact on streamflow, and that is why we included this discussion in our manuscript, but for the purpose of this study we have considered that their inclusion is not necessary.

Lines 445-450:

However, climate variability rather than afforestation is usually the main driver of streamflow reduction (Buendia et al., 2016). To confirm whether in our study this assumption is valid, and it is reasonable to exclude these land use changes without compromising model results, we performed an analysis of trends in model residuals (see section 1 of the Supplement) using the forested area in 2018 for the whole simulation period. This analysis does not evidence the presence of a factor other than the ones already included in the model affecting the hydrological response, and thus justifies the exclusion of land use changes in our study.

I want to acknowledge the effort made by the authors to follow my recommendations. The new information provided is noteworthy but not easy to understand, so I am trying to analyse it and to provide updated recommendations to the authors.

Response: Thank you very much, we have done our best to incorporate these new recommendations and further improve our manuscript.

Trends: The data shown in the table S1 are really striking as they point to relevant internal inconsistencies in the model results. First, the fact that 70% of the gauging stations show positive or negative significant residual trends, with a coefficient of variation of 822%, demonstrates a high uncertainty of the model results. Second, the spatial distribution of these trends in figure S1 shows disordered patterns; some successive gauging stations without any significant tributary between them show opposite trends, such as Castellbell and Abrera stations on the Llobregat River and the two Les Masies de Roda stations on the Ter River. Third, some of the plots in figure S2 show very asymmetric abnormal shapes, either positive (Balsareny, Fogars de la Selva (Pont Eiffel), Sallent) or negative (Guixers, Sant Feliu de Buixalleu).

Response: Thank you for your comment. However, we disagree that these results point to an inconsistent model performance. Despite most trends being statistically significant, their small $R^2$ and Figure S2 evidence that most stations do not present any clear tendency. Moreover, the ones that do – for example, Guixers (Cardener – Monegal) and Vilanova de Sau – have a limited number of observations, and thus the trend is not representative of the entire simulated period. The unequal distribution of observed streamflow can also explain the opposite residual trends between successive gauging stations, as coincidentally the Abrera and Les Masies de Roda (Ter) stations have observations for only half the simulation period, while Castellbell and Les Masies de Roda (Ter i Gurri) have a more complete record. Also, in relation to this and as per the suggestion of Reviewer 2, we have also computed the increase in forested area in the actual drainage area of each of the 50 gauging stations (Table S1), and added to the discussion.

Lastly, the "asymmetric abnormal shapes" in Figure S2 are due to the daily model residuals in $m^3$/s being shown, and therefore the plot is skewed towards the larger residuals during peak flows. This only indicates that some gauging stations tend to overestimate (e.g., Sant Feliu de Buixalleu) or underestimate (e.g., Balsareny) peaks, which does point to worse model performance at these stations, but we do have to consider that there are no perfect models and the fact that we are using so many gauging stations means that it is not possible to find a very good fit for all of them.

Lines 8-15 of the Supplement:

However, we must consider that the increase in forested area varies locally, and not all gauging stations have observed data for the whole period (2001-2022). Therefore, we have used the Corine Land Cover maps from 2000, 2006, 2012 and 2018 (EEA, 2000, 2006, 2012, 2018) to determine for each gauging station the increase in forested area within their drainage area during the closest period to the actual observations (Table S1). Of the 50 gauging stations, most (74%) present an increase in forested area, consistent with the general increase for the whole study area, while 14% present a decrease, and the remaining 12% either do not present any change in forested area or it could not be calculated because the observations start in 2018. Therefore, if the hypothesis that these land use changes are a main driver for streamflow reduction is true, we would expect to observe decreasing trends in model residuals for the gauging stations with an increase in forested area, and vice versa.

Lines 24-34 of the Supplement:

However, of the 37 gauging stations with an increase in forested area, only 54% show a decreasing trend in model residuals (55.6% when only considering statistically significant trends). Similarly, of the 7 gauging stations with a decrease in forested area, 4 show an

increasing trend in model residuals, but only 2 are significant. In summary, of the 44 gauging stations where a change in forested area is observed, only 17 (38.6%) present a trend in model residuals consistent with the change. Moreover, we also must consider that $R^2$ is very small for all trends, with a maximum of 0.098, although 82% of trends present values of $R^2 < 0.01$. Therefore, as we do not observe clear trends in model residuals, we can reasonably assume that land use changes in our study are not a main driver influencing streamflow and their omission for the purpose of our analysis is not incorrect.

Table S1. Trends in the model residuals and change in forested area in the catchment area of each gauging station. Significant trends are marked in bold (p-value < 0.05). LR: Linear Regression. The units for LR slopes are m3/s/day.

| Gauging station | Change in forested area (%) | LR slope | LR p-value | $R^2$ | Likelihood |
|---|---|---|---|---|---|
| Abrera | +16.13 | -7.43E-04 | **5.55E-04** | 3.21E-03 | virtually certain |
| Balsareny | +11.60 | 2.74E-04 | **3.49E-18** | 9.55E-03 | virtually certain |
| Berga | +0.71 | 2.25E-04 | **4.21E-07** | 4.91E-03 | virtually certain |
| Cardona | -1.27 | 2.00E-04 | **1.14E-17** | 1.08E-02 | virtually certain |
| Castellar de n'Hug | +10.26 | 1.60E-05 | **2.71E-02** | 7.93E-04 | very likely |
| Castellbell i el Vilar | +16.05 | 5.04E-04 | **3.69E-16** | 8.31E-03 | virtually certain |
| Castellbisbal | -1.59 | 2.25E-04 | 1.93E-01 | 5.58E-04 | likely |
| Castellet i la Gornal | +16.89 | -1.26E-05 | **3.39E-03** | 2.08E-03 | virtually certain |
| El Papiol | +7.26 | 3.68E-05 | **4.20E-03** | 2.10E-03 | virtually certain |
| Esponellà | +8.09 | -3.20E-04 | **5.70E-12** | 6.00E-03 | virtually certain |
| Fogars de la Selva (Can Simó) | +4.99 | -2.07E-04 | **1.64E-15** | 8.84E-03 | virtually certain |
| Fogars de la Selva (Pont Eiffel) | +2.05 | -1.05E-04 | **2.22E-05** | 2.42E-03 | virtually certain |
| Girona (Onyar) | +19.66 | -1.65E-06 | 9.35E-01 | 8.18E-07 | very unlikely |
| Girona (Ter) | +8.64 | -1.10E-04 | 3.96E-01 | 9.36E-05 | about as likely as not |
| Guardiola de Berguedà | +14.72 | 1.42E-04 | **1.96E-07** | 3.47E-03 | virtually certain |
| Guixers (Aigua de Valls) | 0 | 8.39E-04 | **7.85E-23** | 2.19E-02 | virtually certain |
| Guixers (Cardener - Monegal) | - | -7.24E-04 | **1.38E-34** | 9.80E-02 | virtually certain |
| Jorba | +20.84 | -1.08E-05 | 6.43E-02 | 9.39E-04 | very likely |
| La Cellera de Ter | +8.71 | 2.24E-04 | 1.35E-01 | 3.39E-04 | likely |
| La Coma i la Pedra | +32.51 | -2.33E-04 | **2.05E-11** | 1.73E-02 | virtually certain |
| La Garriga | +9.62 | 3.55E-05 | **6.73E-09** | 4.19E-03 | virtually certain |
| La Pobla de Claramunt | -0.37 | -2.85E-05 | **4.83E-09** | 8.48E-03 | virtually certain |
| Les Masies de Roda (Ter i Gurri) | +8.05 | 3.38E-04 | **1.42E-05** | 2.95E-03 | virtually certain |
| Les Masies de Roda (Ter) | -0.13 | -4.37E-04 | **4.90E-02** | 1.26E-03 | very likely |
| Martorell | -0.26 | -9.67E-06 | 8.40E-01 | 9.89E-06 | unlikely |
| Montornès del Vallès | +6.65 | -1.25E-05 | **9.17E-03** | 9.57E-04 | virtually certain |
| Montseny | +20.40 | -3.41E-06 | 6.52E-01 | 3.57E-05 | about as likely as not |
| Navès | +10.24 | 1.98E-05 | 7.38E-02 | 6.49E-04 | very likely |
| Olot | +6.48 | -7.09E-05 | **1.27E-14** | 7.67E-03 | virtually certain |
| Puig-reig | -3.58 | 2.04E-05 | 1.76E-01 | 4.04E-04 | likely |
| Ripoll | +7.43 | 9.11E-05 | 6.15E-02 | 4.35E-04 | very likely |
| Riudellots de la Selva | +11.10 | -1.91E-04 | **2.43E-34** | 2.40E-02 | virtually certain |
| Sallent | +14.88 | 3.81E-05 | **4.14E-11** | 6.75E-03 | virtually certain |
| Sant Celoni | +12.71 | -2.90E-06 | 6.22E-01 | 3.08E-05 | about as likely as not |
| Sant Feliu de Buixalleu | +9.10 | 4.65E-05 | **7.89E-03** | 1.91E-03 | virtually certain |

| | | | | | |
|---|---|---|---|---|---|
| Sant Gregori | +5.72 | -4.30E-05 | **3.03E-05** | 3.13E-03 | virtually certain |
| Sant Joan de les Abadesses | +9.36 | 5.34E-05 | **2.58E-02** | 6.28E-04 | very likely |
| Sant Joan Despí | +14.78 | 5.75E-04 | **9.37E-08** | 3.73E-03 | virtually certain |
| Sant Sadurní d'Anoia | +14.53 | 2.43E-05 | 9.73E-02 | 3.54E-04 | very likely |
| Sant Vicenç dels Horts | -1.48 | 1.56E-03 | **7.14E-20** | 1.38E-02 | virtually certain |
| Santa Coloma de Gramenet | +5.48 | -1.17E-04 | **5.37E-06** | 2.84E-03 | virtually certain |
| Santa Cristina d'Aro | +36.08 | 3.14E-06 | 8.01E-02 | 5.22E-04 | very likely |
| Santa Perpètua de Mogoda | +2.90 | -1.04E-04 | **1.10E-24** | 2.79E-02 | virtually certain |
| Serra de Daró | +11.39 | -8.87E-05 | **7.51E-03** | 1.09E-03 | virtually certain |
| Torelló | 0 | 1.21E-04 | **2.92E-02** | 1.62E-03 | very likely |
| Torroella de Montgrí | +9.54 | -4.62E-04 | **9.68E-03** | 8.90E-04 | virtually certain |
| Tortellà | +0.69 | -3.08E-04 | **1.94E-06** | 1.05E-02 | virtually certain |
| Vilada (Merdançol) | 0 | -3.75E-06 | 4.59E-01 | 2.02E-04 | about as likely as not |
| Vilada (Riera Vilada) | 0 | 2.36E-05 | 4.87E-01 | 1.41E-04 | about as likely as not |
| Vilanova de Sau | - | 1.00E-03 | **9.33E-13** | 5.01E-02 | virtually certain |

Negative trends of the residuals may be attributed to the role of increased forest cover in the area, a very likely behaviour already demonstrated by previous works, but here it is more difficult to attribute positive trends to hydrological reasons.

Response: As mentioned above and in section 1 of the Supplement, we disagree that the residual trends, despite many being statistically significant, can be attributed to the increase of forest cover or other hydrological factors. Instead, the clear lack of trends consistent to land use changes evidence that these (which are the only potentially main factor affecting streamflow that we do not already include in our model) are not in fact a main factor in the overall study region. This confirms our hypothesis and allows us to proceed with the analysis of hydrological indicators with the model as it is. We reiterate again that this does not mean that the model is a perfect fit, but that it is an acceptable approximation which allows us to perform a more comprehensive analysis of spatio-temporal trends and patterns that we would with the gauging stations available.

I wonder whether the calibration-validation strategy used (randomly selected for each station independently) may lead to different sensitivities of the model to climate forcing at each station in such a varied climate and induce this scatter. As Buendia et al (2016) stated "precipitation appears to follow a generalised decreasing trend, although the significance of these results depended strongly on the time period considered". In fact, the authors do not provide any evidence of the validity of their consideration (line 371): "We consider that, given the high number of gauging stations in our study, randomly determining the calibration and validation periods for each station effectively captures all spatio-temporal variability in both periods. Thus, a bootstrapping method to repeatedly resample the calibration and validation periods, which is time consuming and would imply running many more iterations, is not necessary".

Response: Thank you for your comment. We do not believe that the calibration-validation strategy leads to different sensitivities to climate forcing at specific gauging stations. While it is true that some station might not capture all its climatic variability within its calibration period (e.g., calibration falls during a prolonged drought), other gauging stations within the basin will

compensate, and due to the parameter calibration being basin-wide, for the particular gauging station the model will still be able to simulate climatic conditions not fully captured in the calibration period.

This is the reason for our consideration in line 371 (now line 375), so we have added to it to make it clearer. However, we agree that we do not provide any evidence of that beyond this reasoning. Originally, we started working on a different paper where we tested this calibration-validation strategy among many others, but unfortunately the main researcher working on this exercise left the project before it could be completed. Nonetheless, we already include this discussion in lines 381-388.

Lines 377-379:

In other words, even if the random selection for a specific gauging station leads to a calibration/validation period which does not capture all climatic variability, this variability will be included in other gauging stations, and due to the calibration process being at the basin scale, the model will still account for all variability.

SWAT+: This is not a physically-based model even if it can provide with good results. This qualification of this model contributes to the degradation and loss of usefulness of terminology and concepts. The addition of complementary processes does not modify its essentially empirical character. There is an agreed methodological caveat that just because a model gives good results does not imply that it is for the good reasons (in particular, structure, internal stores and flows). In fact, the results shown by the authors in table S1 may provide a corollary of this principle: despite acceptable flow calibration/validation tests, residual discharge trends show chaotic values difficult to attribute to hydrological reasons.

Response: We have removed the descriptor "physically-based".

Lines 125-127:

SWAT is a semi-distributed ecohydrological model widely used worldwide (Abbaspour et al., 2015; Gassman et al., 2014; Samimi et al., 2020), including many applications in Mediterranean basins (Boithias et al., 2017; Brouziyne et al., 2021; De Girolamo et al., 2022).

Recommendations:

In general terms, the manuscript describes the modelling exercise, shows its results, makes some comparisons with observed data and claims the success of the exercise as it "led to successfully simulating hydrological and anthropogenic processes in water-scarce Mediterranean basins" and "resulted in notable improvements in hydrological modelling and its potential use to support decision-making in the water management sector" without contributing no evidence of these successes and improvements.

Response: Regarding the first claim, we believe that the satisfactory values of the objective functions achieved after calibration and validation are enough evidence of having successfully simulated hydrological and anthropogenic processes in several water-scarce Mediterranean

basins. However, regarding the second claim, we agree that we do not properly quantify the improvements in hydrological modelling, so we have reformulated this sentence.

Lines 533-539:

The spatio-temporal analysis of streamflow patterns and trends have provided insights into the evolution of hydrological dynamics under climate change and increasing anthropogenic pressures in basins vulnerable to water scarcity. Moreover, the integration of first-hand expert knowledge from water managers into our modelling framework has resulted in a more realistic simulation of anthropogenic process, highlighting the potential use of this model to support decision-making in the water management sector. Lastly, the introduction of a randomised calibration and validation approach allows us to overcome the limitations and biases arising of conventional approaches when dealing with multiple gauging stations of variable length without the need for complex analysis.

The authors should not claim good modelling results beyond acceptable tests of efficiency and uncertainty, but should be much more analytical by discussing their strengths and weaknesses and suggesting ways to remedy the latter.

Response: We disagree that we should not claim good modelling results beyond acceptable tests of efficiency and uncertainty, because this is in fact the aim of these tests. We agree of course that any model can be subject to improvements so that it can more closely represent the real system, but for the scope of this study, our model results are proven to be sufficient. In the discussion (mainly section 4.1 but also throughout the other sections) we already underline the weaknesses arising from our approach as well as strategies to address them in future studies.

- The authors cannot justify the reason for the omission of the role of land cover in the hydrological changes of this area on the basis of any published work, nor justify the validity of this omission on the modelling results which are largely inconsistent in this respect.

Response: As per a previous response, we have reformulated part of the text, so it is more clear that we are not omitting the role of land use cover in hydrological changes on the basis of a specific published work. However, we disagree that we cannot justify this omission on our modelling results (section 1 of the Supplement).

Lines 445-450:

However, climate variability rather than afforestation is usually the main driver of streamflow reduction (Buendia et al., 2016). To confirm whether in our study this assumption is valid, and it is reasonable to exclude these land use changes without compromising model results, we performed an analysis of trends in model residuals (see section 1 of the Supplement) using the forested area in 2018 for the whole simulation period. This analysis does not evidence the presence of a factor other than the ones already included in the model affecting the hydrological response, and thus justifies the exclusion of land use changes in our study.

- Following Bieger et al (2017), the authors can claim that SWAT is "one of the most widely used hydrologic models in the world" but cannot claim that it is a physically-based model.

Response: We have removed the descriptor "physically-based".

Lines 125-127:

SWAT is a semi-distributed ecohydrological model widely used worldwide (Abbaspour et al., 2015; Gassman et al., 2014; Samimi et al., 2020), including many applications in Mediterranean basins (Boithias et al., 2017; Brouziyne et al., 2021; De Girolamo et al., 2022).

The model parameters were optimized to obtain the best simulation of discharges, but not the various 'hydrological indicators' extensively exposed in the manuscript. Therefore, due to the equifinality problem (Beven, 2006; Kirchner, 2006), various sets of model parameter may give discharge efficiencies very similar to the best one, but may give quite divergent values of these 'hydrological indicators'.

Response: Thank you for your comment, we agree that the equifinality problem inherent to most hydrological models may result in different values for the hydrological indicators. Therefore, we have also calculated the hydrological indicators using the upper and lower limits of the 95PPU uncertainty bands, so that we can propagate this uncertainty to the Sen's slopes. We visualize this uncertainty in Figure S10 (Section 3 of the Supplement), and we have added the following discussion:

Lines 81-84 of the Supplement:

Figure S9 shows the observed and simulated Sen's slopes of each of the 40 indicators, while Fig. S10 shows only the indicators for which both the observed and simulated trend are significant (Mann-Kendall, p-value < 0.05), as well as the uncertainty associated to the simulated Sen's slopes due to model equifinality. While some of the significant pairs present different directions, the majority are both either positive or negative, even considering the uncertainty.

Lines 92-96 of the Supplement:

[Figure]

Figure S10: Ratio of Sen's slope to mean indicator value only for significant pairs of observed and simulated trends. Uncertainty associated to simulated Sen's slope is also shown. For indicators 18 and 20 in the Fogars de la Selva plot, the Sen's slope is not divided by the indicator value. This is because due to the Sen's slope being very close to 0, the uncertainty value became too large when standardizing, and so it masked the other indicators. See Table S3 to match each indicator to the number used in the figure.

Consequently, model-simulated 'hydrological indicators' face to two severe uncertainties: the role of land cover changes and the issue of model equifinality. It is not possible to determine whether the differences between modelled and observed trends of these indicators are due to the role of land cover change or modelling equifinality effects. Therefore, the statement in line 76 of the Supplement regarding trend analyses is not acceptable: "Moreover, the fact that streamflow was first calibrated ensures overall the validity of the analysis".

Response: As mentioned above, based on section 1 of the Supplement we disagree that land cover changes during our study period are a main driver of streamflow reduction. Also, the uncertainty due to model equifinality has been quantified (see above), and therefore we consider that the calibrated streamflow can be used to conduct the analysis. However, we have removed this line because it did not quite fit the rest of the text.

- Throughout the manuscript, the trends are shown as "Sen slopes", but the units are not always shown, especially for the time variable, so the value of the rate of change is not clear if it is per day, month or year.

Response: Thank you for your comment. We have added the units of slopes where it wasn't already specified.

Lines 293-295:

Table 5. Analysis of trends for the annual percentage of dry river segments, mean annual temperature and mean annual precipitation. Significant trends are marked in bold (p-value < 0.05). Slope units are %/year, ºC/year, and mm/year respectively. LR: Linear Regression; MK: Mann-Kendall.

Lines 305-306:

Figure 4: Spatial distribution of Sen's slope (a, units $hm^3$/year) and standardized Sen's slope (b, units $year^{-1}$) for the hydrological indicator total annual flow.

Lines 312-313:

Figure 5: Spatial distribution of standardized Sen's slope (units $year^{-1}$) for the hydrological indicators annual Q50 (a) and Q50 in January (b), in April (c), in July (d), and in October (e), representative of the different seasonal flow patterns.

Lines 328-330:

Figure 6: Spatial distribution of Sen's slope for the hydrological indicators Q90 (a), Q10 (b), number of high and low flow pulses (c-d), and their mean duration (e-f). Sen's slopes for Q90

and Q10 are standardized (a-b, units year$^{-1}$), while the units for the other indicators are number of events/year (c-d) and days/year (e-f).

Figure 8: Spatial distribution of Sen's slope for the hydrological indicators rise rate (a), fall rate (b) and number of flow reversals (c). The units are number of reversals/year for the last indicator, while they are standardized for the rise and fall rates indicators (units year$^{-1}$).

Table S1. Trends in the model residuals and change in forested area in the catchment area of each gauging station. Significant trends are marked in bold (p-value < 0.05). LR: Linear Regression. The units for LR slopes are m$^3$/s/day.

Figure S1: Spatial distribution of trend slopes (units m$^3$/s/day) for model residuals.

Removing the stronger influence of river segments with higher streamflow in the analysis of temporal trends does not seem to me a sound option when the main objective of the study concerns water resources. On the other hand, the comparison between slopes and mean flows (fractions of runoff gained or loss at annual intervals) are convenient to evaluate their importance in terms of water resources.

Response: We believe that for the purpose of this analysis the standardization of Sen's slopes with volume or flow units is well-grounded, as it allows us to better identify areas of the stream network which follow similar trends regardless of magnitude. This can be observed in Figure 4, where without standardization the interpretation of spatial patterns in temporal trends for Total Annual Flow is skewed due to the larger main reaches of the Llobregat and Ter rivers, but with standardization it can be discerned that smaller tributaries can show in fact more notable trends. However, it is true that the non-standardized Sen's slopes are more relevant in terms of considering water resources, so we have added on to the discussion of Figure 4a.

Significant trends in total annual flow show a general decrease, of up to 6 hm$^3$/year, except in the headwaters of the Llobregat basin (Fig. 4). However, this region shows a poorer model adjustment (see discussion on section 4.1), which may compromise the reliability of this result. Figure 4a shows that larger negative trends can be observed along the course of the main rivers, when significant, most notably along the lower course of the Llobregat river but also observed upstream of the reservoirs in the Ter river as well as near the mouths of the Besòs and Tordera rivers. Removing the stronger influence of reaches with higher flows, the standardized Sen's slopes show that the larger relative negative trends can be observed in smaller tributaries (Fig. 4b).

[Figure]

Figure 4: Spatial distribution of Sen's slope (a, units $hm^3$/year) and standardized Sen's slope (b, units $year^{-1}$) for the hydrological indicator total annual flow.

Lines 417-421:

Significant decreasing trends over the CRBD during the first two decades of the 21st century have been identified for medium and high flows (Fig. 5a & 6a), as well as total annual streamflow (Fig. 4). The larger absolute decreasing trends in total annual streamflow, between -2.08 and up to -6.14 $hm^3$/year, are found in the lower Llobregat and in the Ter river upstream of the reservoirs (Fig. 4a). Both of these areas are of notable interest from the water management perspective to supply Barcelona's metropolitan area, the largest urban centre within the CRBD.

- The hydrographs shown in figures 2 and S3 to S8 are very difficult to understand because the diverse plots cover each other. A logarithmic scale of the discharge axis might help to better visualize the plots.

Response: Thank you for your comment. We have changed Figure 2 for monthly hydrographs instead of daily, because even with a logarithmic scale, daily hydrographs in a single figure were not easy to visualise. Instead, the daily hydrographs (with a logarithmic y-axis) can be found in the Supplement (Figures S3-S8, including the 95PPU uncertainty bands).

Lines 273-275:

The comparisons between observed and simulated streamflow for representative gauging stations of the main rivers of the CRBD demonstrate good model performance (Fig. 2, also see section 2 of the Supplement for daily streamflow and 95PPU uncertainty bands).

[Figure]

Figure 1: Observed and simulated monthly streamflow for the period 2001-2022 in the six main rivers of SWAT+CRBD. Individual KGE and PBIAS values for both the calibration and validation periods are also shown.

[Figure]

Figure S3: Observed and simulated daily streamflow, and the 95PPU band for the gauging station in Castellet i la Gornal, in the Foix basin. KGE and PBIAS values for the calibration and validation periods and the R-factor and P-factor are shown.

[Figure]

Figure S4: Observed and simulated daily streamflow, and the 95PPU band for the gauging station in Esponellà, in the Fluvià basin. KGE and PBIAS values for the calibration and validation periods and the R-factor and P-factor are shown.

[Figure]

Figure S5: Observed and simulated daily streamflow, and the 95PPU band for the gauging station in Fogars de la Selva, in the Tordera basin. KGE and PBIAS values for the calibration and validation periods and the R-factor and P-factor are shown.

[Figure]

Figure S6: Observed and simulated daily streamflow, and the 95PPU band for the gauging station in Les Masies de Roda, in the Ter basin. KGE and PBIAS values for the calibration and validation periods and the R-factor and P-factor are shown.

[Figure]

Figure S7: Observed and simulated daily streamflow, and the 95PPU band for the gauging station in Sant Joan Despí, in the Llobregat basin. KGE and PBIAS values for the calibration and validation periods and the R-factor and P-factor are shown.

[Figure]

Figure S8: Observed and simulated daily streamflow, and the 95PPU band for the gauging station in Santa Coloma de Gramenet, in the Besòs basin. KGE and PBIAS values for the calibration and validation periods and the R-factor and P-factor are shown.

- The units for the variables x and y in the equation enclosed in Figure 3 are not stated. This graph seems to mix observed and simulated results, which should be explained.

Response: We have removed the equation from Figure 3, as we already show the linear regression slope and $R^2$ of the percentage of dry river segments in Table 5. We have also clarified what are simulation results (percentage of dry river segments) and what are observations (mean annual temperature and precipitation).

[Figure]

Figure 3: Evolution of simulated annual percentage of river segments that dry at least once a year, as well as observed mean annual rainfall and temperature.

- The analysis of trends in model residuals in section 1 of the Supplement does not justify the validity of excluding land-cover changes in the study, but demonstrates the difficulty of the modelling exercise to provide reliable estimates of the trends.

Response: As mentioned above, we disagree that the analysis of trends in model residuals in section 1 of the Supplement does not justify the omission of land use changes.

- The units of the slopes shown in table S1 are not shown. Both this table and figures S1 and S2 demonstrate the very inconsistent results of the model exercise with respect to these trends. These results must be further discussed and the sentence "we can reasonably assume that land use changes in our study are not a main driver influencing streamflow" should be deleted.

Response: We have added the units of the slopes in Table S1. However, we disagree that Table S1 and Figures S1 and S2 demonstrate inconsistent model results. Section 1 of the Supplement shows that despite seeing statistically significant trends in model residuals for 70% of the gauging stations, this statistically significance is questionable due to small $R^2$ values and the plots of Figure S2. Moreover, many trends are not consistent with the expected impact of omitting land use changes. Therefore, we can assume that land use changes do not play a factor in residuals trends and thus we can use our calibrated model to proceed with the analysis of trends in hydrological indicators, which was the aim of this exercise.

Table S1. Trends in the model residuals and change in forested area in the catchment area of each gauging station. Significant trends are marked in bold (p-value < 0.05). LR: Linear Regression. The units for LR slopes are $m^3$/s/day.

- The Y-axes in figures S9 and S10 are not appropriate because variables of diverse ranks are shown together. The ratio of slope to mean value (%) could be better as Y-axe units, as this could increase the visibility of low values and allow direct comparison between gauging stations because the axes could be equal.

Response: Thank you for your comment, we have changed the Y-axis units in Figures S9 and S10 to increase visibility of smaller values (and therefore we have excluded Figure S10b). The Y-axis is still different for each gauging stations, but the objective of these figures is only to show if trends have the same direction (negative or positive) using observed and simulated discharge, and not to compare between gauging stations.

Lines 89-96 of the Supplement:

[Figure]

Figure S9: Ratio of Sen's slope to mean indicator value for observed and simulated trends. See Table S3 to match each indicator to the number used in the figure.

[Figure]

Figure S10: Ratio of Sen's slope to mean indicator value only for significant pairs of observed and simulated trends. Uncertainty associated to simulated Sen's slope is also shown. For indicators 18 and 20 in the Fogars de la Selva plot, the Sen's slope is not divided by the indicator value. This is because due to the Sen's slope being very close to 0, the uncertainty value became too large when standardizing, and so it masked the other indicators. See Table S3 to match each indicator to the number used in the figure.

**Reviewer 1**

The authors have taken on the comments and suggestions made in the first review concerning the reformulation of objectives and the definition of end-user input. The supplements are interesting and useful.

In my opinion, the manuscript is acceptable in this 2nd version.

Just one last comment:

The residue study in supplement 1 is a good idea. Trends are presented basin by basin (which is interesting). But these trends are only compared with the percentage increase in forest area over the entire study area. It would have been more convincing to show the variation in forest area (and/or urbanized area) in each basin beside the residual trends.

Response: Thank you very much for your review and your suggestion to improve the analysis of trends in model residuals. We have calculated for each of the 50 gauging stations the actual forest increase within their catchment area while also accounting for the period for which we have observed discharge, as some gauging stations are not representative of the whole 2001-2022 simulated period.

Lines 8-15 of the Supplement:

However, we must consider that the increase in forested area varies locally, and not all gauging stations have observed data for the whole period (2001-2022). Therefore, we have used the Corine Land Cover maps from 2000, 2006, 2012 and 2018 (EEA, 2000, 2006, 2012, 2018) to determine for each gauging station the increase in forested area within their drainage area during the closest period to the actual observations (Table S1). Of the 50 gauging stations, most (74%) present an increase in forested area, consistent with the general increase for the whole study area, while 14% present a decrease, and the remaining 12% either do not present any change in forested area or it could not be calculated because the observations start in 2018. Therefore, if the hypothesis that these land use changes are a main driver for streamflow reduction is true, we would expect to observe decreasing trends in model residuals for the gauging stations with an increase in forested area, and vice versa.

Lines 24-34 of the Supplement:

However, of the 37 gauging stations with an increase in forested area, only 54% show a decreasing trend in model residuals (55.6% when only considering statistically significant trends). Similarly, of the 7 gauging stations with a decrease in forested area, 4 show an increasing trend in model residuals, but only 2 are significant. In summary, of the 44 gauging stations where a change in forested area is observed, only 17 (38.6%) present a trend in model residuals consistent with the change. Moreover, we also must consider that $R^2$ is very small for all trends, with a maximum of 0.098, although 82% of trends present values of $R^2 < 0.01$. Therefore, as we do not observe clear trends in model residuals, we can reasonably assume that land use changes in our study are not a main driver influencing streamflow and their omission for the purpose of our analysis is not incorrect.

Table S1. Trends in the model residuals and change in forested area in the catchment area of each gauging station. Significant trends are marked in bold (p-value < 0.05). LR: Linear Regression. The units for LR slopes are m3/s/day.

| Gauging station | Change in forested area (%) | LR slope | LR p-value | $R^2$ | Likelihood |
|---|---|---|---|---|---|
| Abrera | +16.13 | -7.43E-04 | **5.55E-04** | 3.21E-03 | virtually certain |
| Balsareny | +11.60 | 2.74E-04 | **3.49E-18** | 9.55E-03 | virtually certain |
| Berga | +0.71 | 2.25E-04 | **4.21E-07** | 4.91E-03 | virtually certain |
| Cardona | -1.27 | 2.00E-04 | **1.14E-17** | 1.08E-02 | virtually certain |
| Castellar de n'Hug | +10.26 | 1.60E-05 | **2.71E-02** | 7.93E-04 | very likely |
| Castellbell i el Vilar | +16.05 | 5.04E-04 | **3.69E-16** | 8.31E-03 | virtually certain |
| Castellbisbal | -1.59 | 2.25E-04 | 1.93E-01 | 5.58E-04 | likely |
| Castellet i la Gornal | +16.89 | -1.26E-05 | **3.39E-03** | 2.08E-03 | virtually certain |
| El Papiol | +7.26 | 3.68E-05 | **4.20E-03** | 2.10E-03 | virtually certain |
| Esponellà | +8.09 | -3.20E-04 | **5.70E-12** | 6.00E-03 | virtually certain |
| Fogars de la Selva (Can Simó) | +4.99 | -2.07E-04 | **1.64E-15** | 8.84E-03 | virtually certain |
| Fogars de la Selva (Pont Eiffel) | +2.05 | -1.05E-04 | **2.22E-05** | 2.42E-03 | virtually certain |
| Girona (Onyar) | +19.66 | -1.65E-06 | 9.35E-01 | 8.18E-07 | very unlikely |
| Girona (Ter) | +8.64 | -1.10E-04 | 3.96E-01 | 9.36E-05 | about as likely as not |
| Guardiola de Berguedà | +14.72 | 1.42E-04 | **1.96E-07** | 3.47E-03 | virtually certain |
| Guixers (Aigua de Valls) | 0 | 8.39E-04 | **7.85E-23** | 2.19E-02 | virtually certain |
| Guixers (Cardener - Monegal) | - | -7.24E-04 | **1.38E-34** | 9.80E-02 | virtually certain |
| Jorba | +20.84 | -1.08E-05 | 6.43E-02 | 9.39E-04 | very likely |
| La Cellera de Ter | +8.71 | 2.24E-04 | 1.35E-01 | 3.39E-04 | likely |
| La Coma i la Pedra | +32.51 | -2.33E-04 | **2.05E-11** | 1.73E-02 | virtually certain |
| La Garriga | +9.62 | 3.55E-05 | **6.73E-09** | 4.19E-03 | virtually certain |
| La Pobla de Claramunt | -0.37 | -2.85E-05 | **4.83E-09** | 8.48E-03 | virtually certain |
| Les Masies de Roda (Ter i Gurri) | +8.05 | 3.38E-04 | **1.42E-05** | 2.95E-03 | virtually certain |
| Les Masies de Roda (Ter) | -0.13 | -4.37E-04 | **4.90E-02** | 1.26E-03 | very likely |
| Martorell | -0.26 | -9.67E-06 | 8.40E-01 | 9.89E-06 | unlikely |
| Montornès del Vallès | +6.65 | -1.25E-05 | **9.17E-03** | 9.57E-04 | virtually certain |
| Montseny | +20.40 | -3.41E-06 | 6.52E-01 | 3.57E-05 | about as likely as not |
| Navès | +10.24 | 1.98E-05 | 7.38E-02 | 6.49E-04 | very likely |
| Olot | +6.48 | -7.09E-05 | **1.27E-14** | 7.67E-03 | virtually certain |
| Puig-reig | -3.58 | 2.04E-05 | 1.76E-01 | 4.04E-04 | likely |
| Ripoll | +7.43 | 9.11E-05 | 6.15E-02 | 4.35E-04 | very likely |
| Riudellots de la Selva | +11.10 | -1.91E-04 | **2.43E-34** | 2.40E-02 | virtually certain |
| Sallent | +14.88 | 3.81E-05 | **4.14E-11** | 6.75E-03 | virtually certain |
| Sant Celoni | +12.71 | -2.90E-06 | 6.22E-01 | 3.08E-05 | about as likely as not |
| Sant Feliu de Buixalleu | +9.10 | 4.65E-05 | **7.89E-03** | 1.91E-03 | virtually certain |
| Sant Gregori | +5.72 | -4.30E-05 | **3.03E-05** | 3.13E-03 | virtually certain |
| Sant Joan de les Abadesses | +9.36 | 5.34E-05 | **2.58E-02** | 6.28E-04 | very likely |
| Sant Joan Despí | +14.78 | 5.75E-04 | **9.37E-08** | 3.73E-03 | virtually certain |
| Sant Sadurní d'Anoia | +14.53 | 2.43E-05 | 9.73E-02 | 3.54E-04 | very likely |
| Sant Vicenç dels Horts | -1.48 | 1.56E-03 | **7.14E-20** | 1.38E-02 | virtually certain |
| Santa Coloma de Gramenet | +5.48 | -1.17E-04 | **5.37E-06** | 2.84E-03 | virtually certain |
| Santa Cristina d'Aro | +36.08 | 3.14E-06 | 8.01E-02 | 5.22E-04 | very likely |
| Santa Perpètua de Mogoda | +2.90 | -1.04E-04 | **1.10E-24** | 2.79E-02 | virtually certain |
| Serra de Daró | +11.39 | -8.87E-05 | **7.51E-03** | 1.09E-03 | virtually certain |
| Torelló | 0 | 1.21E-04 | **2.92E-02** | 1.62E-03 | very likely |
| Torroella de Montgrí | +9.54 | -4.62E-04 | **9.68E-03** | 8.90E-04 | virtually certain |
| Tortellà | +0.69 | -3.08E-04 | **1.94E-06** | 1.05E-02 | virtually certain |

| | | | | | |
|---|---|---|---|---|---|
| Vilada (Merdançol) | 0 | -3.75E-06 | 4.59E-01 | 2.02E-04 | about as likely as not |
| Vilada (Riera Vilada) | 0 | 2.36E-05 | 4.87E-01 | 1.41E-04 | about as likely as not |
| Vilanova de Sau | - | 1.00E-03 | **9.33E-13** | 5.01E-02 | virtually certain |